# High-temperature electrothermal remediation of multi-pollutants in soil

Bing Deng [1,10] ✉, Robert A. Carter[1,10], Yi Cheng [1,10], Yuan Liu[2], Lucas Eddy [1,3,4], Kevin M. Wyss[1], Mine G. Ucak-Astarlioglu[5], Duy Xuan Luong[1,3], Xiaodong Gao[6,7], Khalil JeBailey[8], Carter Kittrell[1], Shichen Xu[1], Debadrita Jana[6], Mark Albert Torres[6], Janet Braam [2] & James M. Tour [1,4,8,9] ✉

Soil contamination is an environmental issue due to increasing anthropogenic activities. Existing processes for soil remediation suffer from long treatment time and lack generality because of different sources, occurrences, and properties of pollutants. Here, we report a high-temperature electrothermal process for rapid, water-free remediation of multiple pollutants in soil. The temperature of contaminated soil with carbon additives ramps up to 1000 to 3000 °C as needed within seconds via pulsed direct current input, enabling the vaporization of heavy metals like Cd, Hg, Pb, Co, Ni, and Cu, and graphitization of persistent organic pollutants like polycyclic aromatic hydrocarbons. The rapid treatment retains soil mineral constituents while increases infiltration rate and exchangeable nutrient supply, leading to soil fertilization and improved germination rates. We propose strategies for upscaling and field applications. Techno-economic analysis indicates the process holds the potential for being more energy-efficient and cost-effective compared to soil washing or thermal desorption.

Soil contamination is a pressing global environmental concern due to the rapid expansion of industrial activities, mining tailings, overuse of agricultural chemicals, and improper waste disposal[1]. Depending on the pollution sources[1], the common contaminants in soil include heavy metals[1,2] including lead (Pb), arsenic (As), zinc (Zn), cobalt (Co), cadmium (Cd), copper (Cu), mercury (Hg), and nickel (Ni), as well as persistent organic pollutants (POP) such as polycyclic aromatic hydrocarbons (PAH)[3], polychlorinated biphenyl[4], organochlorine pesticides[5], and total petroleum hydrocarbons[6]. Soil contamination poses significant risks to both human and ecosystems by damaging the water quality and the food chain[7] and reducing land usability for agriculture[1,8]. Urgent and efficient remediation practices are required to address this issue.

Existing technologies for remediating heavy metal and POP-contaminated soil include thermal desorption[9], immobilization[10], soil washing[11,12], advanced oxidation processes[13], bioremediation[14], and others. While generally applicable, these methods face several challenges. First, their remediation speeds are usually slow due to the intrinsic reaction and diffusion kinetics limits, which cannot keep up with the increasing demand for immediate remediation[15]. Second, multiple approaches are required to address co-contamination of soil by heavy metals and organic contaminants, due to their varied occurrences, speciation, and physical and chemical properties[8]. Even worse, multiple pollutants may interfere or compete, reducing the remediation efficiency[16,17]. For example, highly concentrated heavy metals can inhibit microbial metabolism activities, thereby reducing

[1]Department of Chemistry, Rice University, Houston, TX 77005, USA. [2]Department of BioSciences, Rice University, Houston, TX 77005, USA. [3]Applied Physics Program, Rice University, Houston, TX 77005, USA. [4]Smalley-Curl Institute, Rice University, Houston, TX 77005, USA. [5]Geotechnical and Structures Laboratory, U.S. Army Engineer Research & Development Center, Vicksburg, MS 39180, USA. [6]Department of Earth, Environmental, & Planetary Sciences, Rice University, Houston, TX 77005, USA. [7]Carbon Hub, Rice University, Houston, TX 77005, USA. [8]Department of Materials Science and NanoEngineering, Rice University, Houston, TX 77005, USA. [9]NanoCarbon Center and the Rice Advanced Materials Institute, Rice University, Houston, TX 77005, USA. [10]These authors contributed equally: Bing Deng, Robert A. Carter, Yi Cheng. ✉e-mail: bingdeng@rice.edu; tour@rice.edu

the degradation efficiency of organic pollutants[16]. This necessitates highly versatile remediation methods that can effectively address the coexistence of multiple pollutants[18], given the increasing occurrence of co-contaminated soils[14,19]. Only a few methods exist that can simultaneously remove heavy metals and POP, such as the photocatalysis process[20,21], which removes the organics through oxidation and the metal ions through reduction immobilization. Third, some soil remediation approaches require high consumption of chemicals and generate large wastewater streams[11,12], which can burden the economics and lead to secondary pollution.

Recently, electric heating has emerged as a rapid, energy-efficient thermal treatment process for materials production[22] and waste management[23]. By designing the direct Joule heating process, metal nanoparticles[24] and high-entropy alloy nanoparticles[25] were synthesized through thermal shock, which has found widespread application in the production of functional materials for energy storage[26,27] and catalysis[28,29]. Our group developed the flash Joule heating method for converting carbon resources into graphene materials[30]. Furthermore, the flash Joule heating process has been extended to include waste management applications such as plastic upcycling[31], critical metals recovery[32–34], and battery recycling[35,36].

Here, we present a high-temperature electrothermal process (HET) for the effective remediation of multiple pollutants in contaminated soil. By incorporating carbon conductive additives such as environmentally friendly biochar, the soil temperature can be rapidly increased to 1000 to 3000 °C as needed with a heating rate of ~$10^4$ °C s$^{-1}$ using pulsed electric input, followed by a rapid cooling rate of ~$10^3$ °C s$^{-1}$. This high temperature allows for the removal of toxic heavy metals, including Cd, Hg, Pb, Co, Ni, and Cu, which are vaporized and reduced to below regulatory levels. Simultaneously, persistent organic pollutants such as PAH are graphitized, thereby being stable and nontoxic. Due to the ultrafast processing time, the change of soil particle size and major mineral composition remain minimal. Notably, the rapid high-temperature treatment regulates some soil properties, including an increased water infiltration rate and enhanced exchangeable nutrient pool by rapid mineralization of soil organic matter, leading to soil fertilization and improved germination rates by 20 to 30%. Unlike conventional thermal processes that rely on heat transfer, HET directs most of its energy to the soil sample, with a low energy consumption of ~420 kWh tonne$^{-1}$. We propose the prototypes for both ex-situ upscaling and on-site field application for deployment. Life-cycle assessment and techno-economic analysis indicate that the HET process could require less energy consumption and operating expense compared to existing soil remediation techniques, such as thermal desorption or soil washing. With its versatility in remediating multiple pollutants, ultrafast operation within seconds to minutes, relatively low energy demand and overall expense, and zero water usage, the HET process would be a harbinger for near-future soil remediation practice.

## Results and discussion
### Concept of the high-temperature electrothermal process for soil remediation
In the HET process, dry soil is mixed with conductive additives to ensure good electrical conductivity. A high-voltage pulse input within seconds controllably brings the soil to a typical temperature of 1000 to 3000 °C as needed (Fig. 1a). At this high temperature, heavy metals are carbothermically reduced and vaporized (Fig. 1b), and the vapors can then be collected via the vapor extraction pipes (Fig. 1a), which are commonly utilized in traditional thermal desorption remediation techniques. The high temperature simultaneously destroys the POP. For instance, PAH are carbonized to graphite (Fig. 1b), the most stable form of carbon, which is a naturally occurring nontoxic mineral[37].

We initially performed a proof-of-concept test for the HET process on a bench scale (Supplementary Fig. 1). In a typical process, contaminated soil (c-Soil) was mixed with appropriate amounts of conductive additive, such as carbon black (CB) and biochar. The c-Soil and CB mixture was loaded into a quartz tube. The resistance of the sample

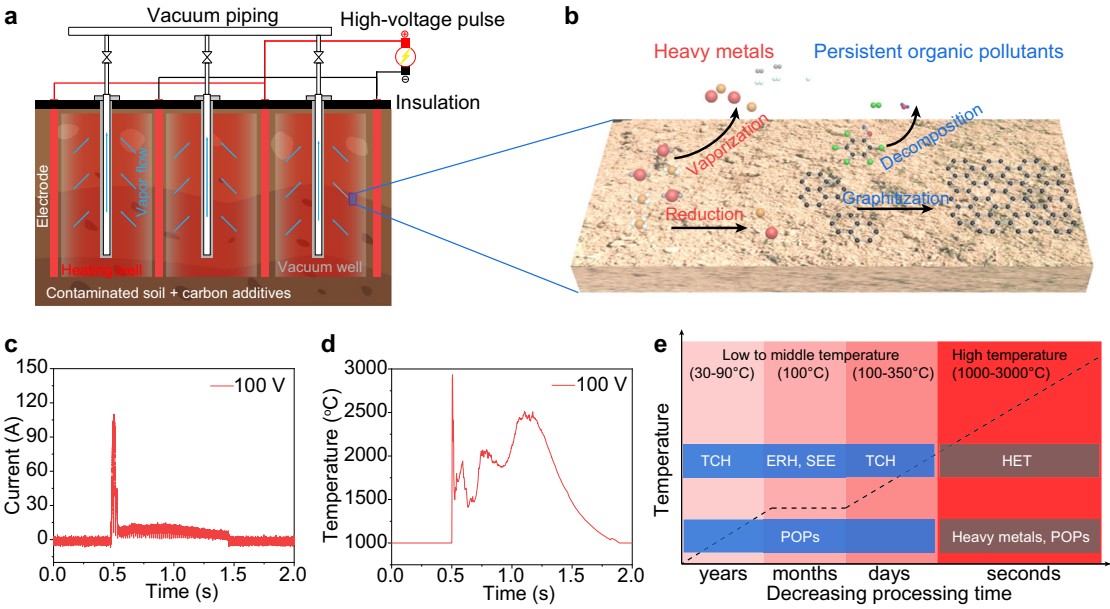

**Fig. 1 | Concept of the high-temperature electrothermal process (HET) for soil remediation. a** Schematic of the HET process, combined with vacuum extraction well. The vacuum piping and insulation blanket remain standard to known thermal remediation methods, but in the case of HET, the electrodes provide a rapid voltage pulse for electric heating, rather than long-duration heat injection. The soil is premixed in place, with biochar or other conductive carbon to provide sufficient conductivity. **b** Schematic showing the removal of heavy metals by reduction and vaporization, and the removal of persistent organic pollutant (POP) by graphitization for PAH. **c** Current curve at an electric input of 100 V for 1 s. **d** Real-time temperature curve of the soil sample at an electric input of 100 V for 1 s. **e** Comparison of the HET with other thermal remediation processes, including thermal conduction heating (TCH), electrical resistance heating (ERH), and steam-enhanced extraction (SEE). While operated at lower temperatures, the latter methods require long treatment periods.

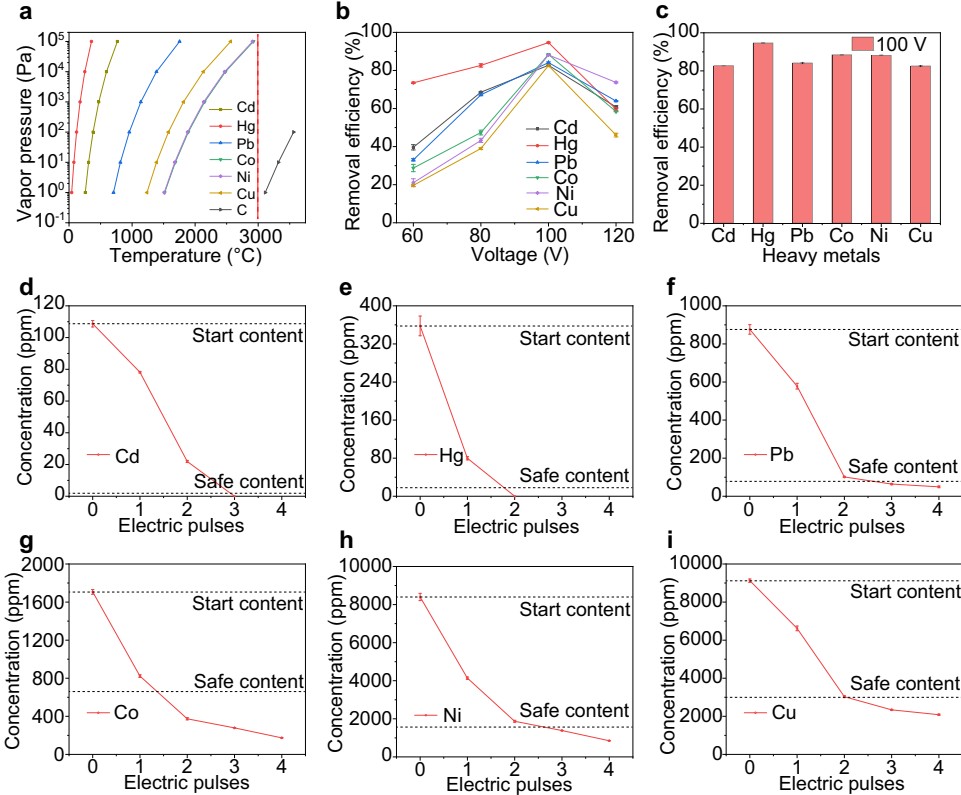

**Fig. 2 | Removal of toxic heavy metals in soil by vaporization. a** Vapor pressure-temperature relationships of representative heavy metals and carbon. The red dashed line denotes the temperature of 3000 °C. **b** The removal efficiencies of representative heavy metals from soil varied with HET voltages. **c** The removal efficiencies of heavy metals at an HET voltage of 100 V. The contents of heavy metals in soil after repetitive electric pulses for (**d**) Cd, (**e**) Hg, (**f**) Pb, (**g**) Co, (**h**) Ni, and (**i**) Cu. The safe contents are from the standard of California Human Health Screening Levels[33]. The concentration of 0 denotes the value lower than the detecting limits of ICP-OES. The error bars denote SD where $N = 3$.

was regulated by compressing the graphite electrodes, which were connected to a capacitor bank with a total capacitance of $C = 60$ mF (Supplementary Fig. 2). At a voltage of $V = 100$ V, discharging time of $t = 1$ s, and sample resistance of $R = 1$ Ω, the current curve showed a maximum value of ~100 A (Fig. 1c). Discharging the capacitor bank brings the sample to a high temperature ranging from 1000 to 3000 °C, depending on the applied voltages (Fig. 1d). The heating and cooling rates were calculated to be ~$1.08 \times 10^4$ °C s⁻¹ and ~$1.88 \times 10^3$ °C s⁻¹, respectively. The ultrafast heating is achieved through pulsed electric input, while the rapid cooling is attributed to efficient heat dissipation resulting from intense thermal radiation (Supplementary Fig. 3). Degassing can occur during the HET process, resulting in resistance variation and temperature oscillation (Fig. 1d).

The HET process is characterized by its high temperature (1000–3000 °C as needed), which is significant higher than that of conventional thermal desorption technologies (usually <400 °C), such as thermal conduction heating (TCH)[38], steam-enhanced extraction (SEE)[39], electrical resistance heating (ERH)[40], and radio frequency heating (RFH)[41]. This feature makes the HET capable of unique performances that are not accessible by traditional processes (Fig. 1e). First, the high temperature enables the simultaneous removal of heavy metals and organic pollutants, whereas thermal desorption methods are only suitable for the remediation of volatile or semi-volatile contaminants[9]. Second, while thermal desorption is a physical process that relies on volatilization as the main mechanism for removing contaminants[9], the HET process destroys organic contaminants in-situ, converting them to naturally occurring nontoxic graphitic minerals. Third, the high temperature of the HET process significantly accelerates both the reaction and diffusion kinetics, allowing remediation to be completed in seconds, which is substantially faster than the low- to

mid-temperature processes that take months or even years to operate[42].

## Removal of toxic heavy metals by reduction and vaporization

The as-collected clean soils possess heavy metal concentrations far below the regulation limits (Supplementary Fig. 4a). Considering the different toxicity, disparate safety standards, and large variation in hazardous levels among real-world contaminated sites for different heavy metals[43], the clean soil sample was co-contaminated by simultaneously spiking with metal salts, primarily metal chlorides: Cd (~100 part per million, ppm), Hg (~300 ppm), Pb (~1000 ppm), Co (~2000 ppm), Ni (~10000 ppm), and Cu (~10000 ppm) (Supplementary Fig. 4b). The concentrations of heavy metals in the c-Soil and the remediated soil (r-Soil) by HET were measured using inductively coupled plasma optical emission spectrometry (ICP-OES) after digestion, and the removal efficiencies were calculated. Most of the heavy metals, including Cd, Hg, Pb, Co, Cu, and Ni, could be vaporized at high temperatures (1000–3000 °C), irrespective of their chemical forms being metal salts or elemental metals (Supplementary Table 1, Fig. 2a).

We investigated the removal efficiencies of heavy metals at different voltages (Fig. 2b, Supplementary Table 2). The removal efficiencies improved from 60 V to 100 V, owing to the higher temperature resulting from a higher voltage (Supplementary Fig. 5). However, excessively high voltages could lead to inhomogeneous heating due to rapid degassing and large sample resistance variation (Supplementary Figs. 5, 6), which slightly reduced the removal efficiencies (Fig. 2b). At the optimized HET voltage of 100 V, the removal efficiencies of all heavy metals were >80% in a single electric pulse (Fig. 2c). The concentrations of heavy metals in CB were much lower

than those in c-Soil (Supplementary Fig. 7), and hence the use of CB as conductive additives would not introduce any significant error.

In addition to CB and biochar, other inexpensive carbon materials with adequate conductivity can also be used as the conductive additives. For example, we demonstrated the applicability of metallurgical coke (Metcoke), flash graphene[30], and even plastic pyrolysis ash, a byproduct of the plastic pyrolysis recycling process[44] (Supplementary Fig. 8). The use of plastic pyrolysis ash is particularly attractive because of its low or even negative value[45], minimizing the materials cost of the HET process. Moreover, the continuously growing plastic waste streams provide an abundant supply for soil remediation purposes. As a result of using carbon conductive additives, a significant amount of residual carbon was left in the r-Soil (Supplementary Fig. 9). Based on the particle size difference between soil and the introduced carbon, it is easy to separate residual carbon from the soil by sieving. By using Metcoke as an example, the separation of the treated soil and residual Metcoke was realized, with the carbon recovery yield of ~92% (Supplementary Fig. 10a–d). The recycled Metcoke was converted into flash graphene (Supplementary Fig. 11), which has better conductivity and can be reused for the HET process (Supplementary Fig. 10e, f), greatly reducing the materials consumption. We measured the soil carbon content in the raw soil and the treated soil after separating carbon conductive additives (Supplementary Fig. 12). The carbon content in the treated soil is ~3.5%, comparable to the raw soil (~3.7%). The residual carbon additive can compensate for the organic carbon loss during the HET process, resulting in a similar total carbon content in the treated soil and raw soil. Other inexpensive carbon additives, such as bituminous activated charcoal, could also be used for the separation (Supplementary Fig. 13).

Unlike some physicochemical adsorption methods that rely on sorbent capacity[10], the HET process for heavy metal removal has no capacity limit. By increasing the number of electric pulses, heavy metal concentration in soil can be continuously reduced. The concentrations of all representative heavy metals were reduced to below California Human Health Screening Levels for residential locales[33] by two to three electric pulses, each lasting only 1 second (Fig. 2d-i). The number of pulses required depends on initial concentrations, safety thresholds, and the vapor pressure of specific heavy metals. We further analyzed the mass balance of heavy metals during the HET process. Using Cu and Ni as examples, X-ray photoelectron spectroscopy (XPS) was conducted to qualitatively determine their distribution. The Ni and Cu peaks were clearly identified for the c-Soil (Supplementary Fig. 14a, b), but no peaks were detectable for the r-Soil, indicating efficient heavy metal removal (Supplementary Fig. 14c, d). Intriguingly, the heavy metals were detected on the quartz tube side walls (Supplementary Fig. 14e, f). Furthermore, we integrated a vacuum system to collect evaporative heavy metals in a trap (Supplementary Fig. 15a), and found that most heavy metals were either evaporated or deposited on the quartz tube (Supplementary Fig. 15b). The evaporated or deposited heavy metals can be captured and properly handled, preventing them from being released into the environment. This process is compatible with vacuum extraction wells to collect the vaporized contaminants used in the traditional thermal remediation processes[46,47].

Contaminated soil containing heavy metals exhibits a wide range of speciation. We analyzed the influence of chemical species on the removal efficiency (Supplementary Note 1). Depending on the thermal properties of the heavy metal species, the elevated temperatures in the HET process can initiate a sequence of reactions, including evaporation, thermal decomposition, and carbothermic reduction. We here considered Hg as an example. Under the HET process (Supplementary Fig. 16a, b), representative Hg compounds such as $HgCl_2$, $HgO$, and $HgSO_4$ can be converted to Hg at temperatures below 1200 °C. By using a single HET pulse, high removal efficiencies were achieved for all tested Hg species (Supplementary Fig. 16c, d): Hg (~90.4%), $HgCl_2$ (~94.6%), $HgO$ (~95.1%), and $HgSO_4$ (~86.5%). This further demonstrates the broad applicability of the HET method for remediating heavy metal-contaminated soil.

## Removal of PAH by graphitization

In addition to heavy metals, the HET process can also destroy POP. To test this, we initially attempted to remediate PAH-contaminated soil using pyrene, fluorene, and benz[a]anthracene as representative compounds. The clean soil was spiked with individual PAH, and then mixed with carbon black as a conductive additive. The HET conditions for PAH remediation are listed in Supplementary Table 2, with a typical maximum temperature of ~1500 °C (Supplementary Fig. 17). The PAH in c-Soil and r-Soil after the HET treatment were extracted into an organic phase using solvent extraction[48], and the PAH concentrations were measured using ultraviolet-visible (UV-Vis) spectrophotometry[49] (Supplementary Fig. 18).

The UV-Vis adsorption spectra of pyrene show two characteristic peaks at ~319 and ~333 nm (Fig. 3a). The intensity of these peaks progressively decreased with increasing electric pulses (Fig. 3a). After 3 electric pulses, the pyrene concentration was reduced to below the preliminary remedial goals (PRG) of 2300 ppm (Ref. 3) (Fig. 3b). Similarly, fluorene exhibits a characteristic adsorption peak at ~299 nm, whose intensity was greatly reduced after HET (Fig. 3c), and to below its PRG of 2700 ppm (Ref. 3) by 3 electric pulses (Fig. 3d). The same strategy applied to the remediation of benz[a]anthracene-contaminated soil (Fig. 3e, f), demonstrating the generality of the process. Benz[a]anthracene has a low PRG of 0.62 ppm (ref. 3), which is beyond the detection limit of UV-Vis spectrophotometry after 3 electric pulses (Fig. 3f). In this case, gas chromatography-mass spectrometry (GC-MS) was used for quantification, with the detection limit down to 0.001 ppm (Supplementary Fig. 19). After 6 electric pulses, the content of benz[a]anthracene was reduced to below its PRG (Fig. 3f). In addition to carbon black, other environmentally friendly carbon sources like biochar could be used as the conductive additives and shows similar effectiveness (Supplementary Fig. 20).

The high temperature can graphitize the carbon-containing precursors, as demonstrated in our previous reports on the synthesis of flash graphene from various carbon sources[30]. The Raman spectra of the PAH-contaminated soil after HET treatment exhibit strong 2D bands (Supplementary Fig. 21), indicating the conversion of the carbon additive and soil organic matter to graphitized carbon. Furthermore, we conducted the mass balance measurement of PAH. We developed an apparatus to collect the evaporated PAH (Supplementary Fig. 22a), which are minor compared to the graphitized portion, clearly demonstrating that PAH are removed by graphitization instead of evaporative loss (Supplementary Fig. 22b-d). While graphitized carbon is naturally occurring and non-toxic[37], its chemical stability greatly retards its microbial decomposition and essentially removes it from the global carbon dioxide cycle[50], contributing to the mitigation of greenhouse gas emission.

## Soil properties after the HET process

The soil properties after the HET treatment were assessed, as they are significant for the soil reuse in agriculture. In this case, the HET process employed for soil treatment was typically carried out at ~1500 °C for 1 s, which aligns with the requirements for PAH removal. Metcoke, serving as the conductive additive, was eliminated through sieving before measuring the soil properties. Firstly, we analyzed the physical properties and mineral constitutes of the soil. The soil particle size distribution was measured using a laser particle size analyzer (Supplementary Fig. 23). The treated soil exhibits a slightly higher sand composition compared to the raw soil, primarily attributed to soil aggregation during the HET process (Fig. 4a). Both the raw soil and treated soil were classified as sandy loam. Scanning electron microscopy (SEM) was used to examine the morphology of the treated soil, which revealed a fine powder feature that resembles the raw soil

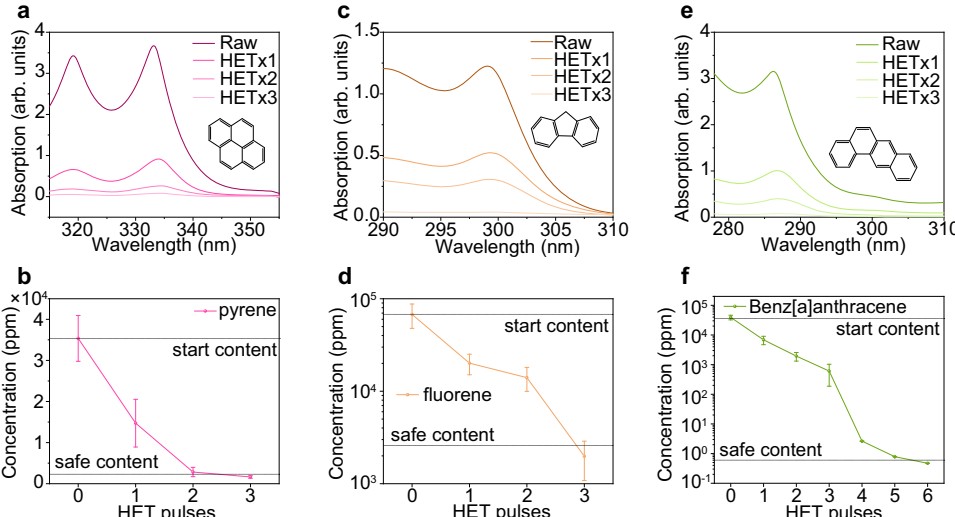

**Fig. 3 | Removal of PAH in soil by graphitization. a** UV-Vis absorption spectra of extracts from pyrene-contaminated soil and r-Soil after repetitive high-temperature electrothermal (HET) pulses. Inset, the chemical structure of pyrene. **b** The content of pyrene in soil varied with repetitive HET pulses. **c** UV-Vis absorption spectra of extracts from fluorene-contaminated soil and r-Soil after repetitive HET pulses. Inset, the chemical structure of fluorene. **d** The content of fluorene in soil varied with repetitive HET pulses. **e** UV-Vis absorption spectra of extracts from benz[a]anthracene-contaminated soil and the r-Soil after repetitive HET pulses. Inset, the chemical structure of benz[a]anthracene. **f** The content of benz[a]anthracene in soil varied with repetitive HET pulses. The safe contents in **b**, **d**, and **f** denote the preliminary remediation goals (4). The error bars in **b**, **d**, and **f** denote the SD where $N = 3$.

(Supplementary Fig. 24). The main crystalline compositions of the raw soil and treated soil were characterized by X-ray diffraction (XRD), where $SiO_2$ and $CaCO_3$ are the major crystal components of the raw soil (Fig. 4b). After the HET process, the $SiO_2$ remained prominent (Fig. 4b), while the calcite was absent, presumably due to its thermal decomposition into calcium oxide. The main composition of the treated soil was further quantified by X-ray fluorescence (XRF), which shows that various oxides underwent almost no change (Fig. 4c). The above analysis shows that, apart from the removal of contaminants, the morphology, particle size, and mineral constituents of the treated soil changed little by the HET process. We attribute this to its ultrafast heating and cooling rates, as well as short heating duration.

Next, we evaluated the water infiltration on the raw soil and treated soil at the bench scale. Water infiltration is crucial for managing agricultural water and replenishing groundwater from runoff[51], where high infiltration rates are typically linked to large, continuous, interconnected macropores[52]. With the same soil volume, the liquid level decreased more quickly in the treated soil than in the raw soil (Fig. 4d, e, Supplementary Fig. 25). The average infiltration rates for the raw soil and treated soil were ~71 cm h$^{-1}$ and ~111 cm h$^{-1}$, respectively (Fig. 4f). The enhanced infiltration rate observed in the HET-treated soil can be ascribed to the increased sand ratio (Fig. 4a, Supplementary Fig. 23), as water tends to flow more rapidly through the large pores in sandy soil compared to the smaller pores in clay soil[53]. Additionally, the presence of residual carbon in HET-treated soil following the sieving process (Supplementary Fig. 12) could contribute to soil porosity and further facilitate water infiltration.

Finally, we conducted plant assays using broccoli sprouts to demonstrate the applicability of the HET method for agricultural land remediation (Fig. 4g). For the plant assay, the treated soil was mixed with raw soil at a weight ratio of 1:1 (denoted as 50% treated soil), and the raw soil was used as the control. The broccoli seeds were regularly watered without additional nutrients. In other pots, vermiculite was mixed into the soils to improve drainage. In both cases, the treated soil showed a 20 to 30% higher germination rates than the raw soil (Fig. 4h), indicating that HET process might enhance plant growth. To explain this, the exchangeable nutrient contents, including P, Ca, K, Mg, Mn, Fe, and nitrate-nitrogen, in the raw and treated soil were

measured (Supplementary Figs. 26, 27). Note that the use of carbon conductive additives in the HET process does not introduce significant error in the measurement (Supplementary Fig. 28). Compared to the raw soil, the exchangeable Fe, P, N, Mn, and Ca in HET-treated soil improved by 4 to 103%, while K and Mg decreased by 16 to 24% (Fig. 4i). The slight decrease in exchangeable K and Mg may be attributed to their higher volatility, leading to the evaporative losses, when compared to other metals like Ca, Fe, and Mn. The dominant increase in exchangeable nutrients can be attributed to the rapid mineralization of soil organic matter by the high-temperature HET process[54,55], contributing to the increased germination rate.

## Upscaling strategies and field application potential

The removal of contaminants relies on the achievable temperature, therefore maintaining a constant temperature is critical when scaling up the HET process. Our analysis shows that the mass per batch could be increased by linearly increasing the voltage or capacitance (Supplementary Note 2). We have upscaled the sample mass to ~8 g per batch, achieving a total treated soil mass of ~100 g with processing time of <10 min (Supplementary Note 2, Supplementary Fig. 29).

In our previous experiments, we utilized capacitors to supply direct current for the HET process (DC-HET). An alternating current (AC) source can also be used for the HET process (AC-HET). To avoid electric overload, the AC-HET system comprises two circuit breakers (Supplementary Fig. 30a, b). The standard AC electricity with a voltage of 120 V and a frequency of 60 Hz was used. The removal efficiencies of different heavy metals were found to be 40 to 80% after a single electric pulse (Supplementary Fig. 30c). This efficiency is slightly lower than that of DC-HET (Fig. 2b), presumably due to the lower temperature of the AC-HET process (Supplementary Fig. 30d). While the temperature is limited by our accessible AC source (120 V) in the laboratory, increasing the voltage would improve the temperature according to our calculations (Supplementary Note 2). The AC-HET process is more suitable for scaling up. By using the AC-HET, we realized the remediation of pyrene-contaminated soil with a mass of 100 g per batch, with a retention time of ~1 min, resulting in a potential production rate of >100 kg day$^{-1}$ in a laboratory scale (Supplementary Fig. 31, Supplementary Note 2). The upscaled sample exhibited

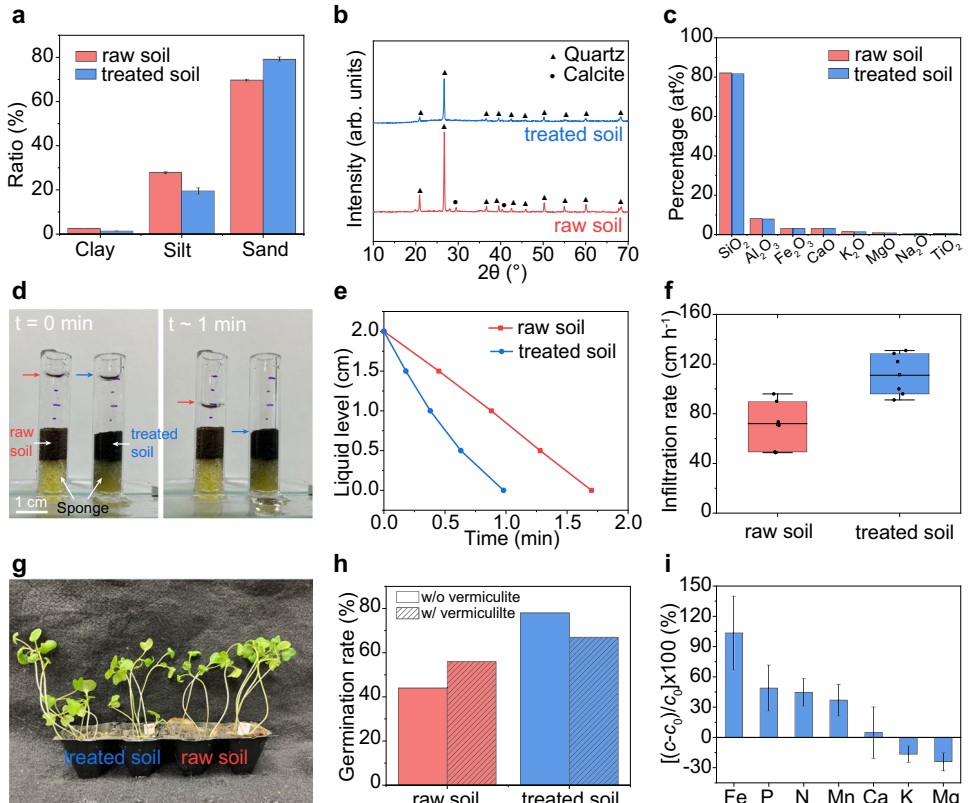

**Fig. 4 | Soil characterization after HET treatment. a** Soil type of raw soil and treated soil based on particle size: clay (<2 μm), silt (2 to 50 μm), and sand (>50 μm). **b** XRD patterns of raw soil and HET-treated soil. The triangle and dot denote quartz and calcite, respectively. **c** XRF-measured percentage of metal oxide composition of raw soil and HET-treated soil. **d** Pictures of water infiltration test at $t = 0$ min (left) and 1 min (right) for raw and HET-treated soil. **e** Liquid level varied with time for raw soil, and HET-treated soil. **f** Boxplot with individual data points of water infiltration rates for raw soil and HET-treated soil; $N = 6$. Center line, median; box limits, upper and lower quartiles; whiskers, 5th and 95th percentiles. **g** Picture of the growth of broccoli on raw soil and 50% treated soil. **h** Gemination rate of broccoli using raw soil and HET-treated soil, with or without vermiculite additive. **i** Exchangeable soil nutrient content change during the HET process. $c_0$ and $c$ are the concentration of nutrients in raw soil and HET-treated soil, respectively. The error bars denote SD where $N = 3$.

comparable pyrene removal efficiency to that of small-scale samples. In industry, high voltage or even ultrahigh voltage technologies are already established[56,57], which could be introduced to further enhance the removal efficiencies and volume per batch.

The HET could possibly be integrated into industrial upscaled techniques such as belt roller for continuous processing (Supplementary Note 2, Supplementary Fig. 32). Commercial scaling of the flash Joule heating process is ongoing, with a production rate of 1 tonne per day at Q2 2023 (ref. 58). The developed equipment and processes could be adapted for the HET soil remediation purposes. Considering that soil removal from remote sites is costly, we also propose the conceptual design of a tractor-attached HET unit (Supplementary Note 2, Supplementary Fig. 33) and a field facility (Fig. 1a, Supplementary Fig. 34) for on-site remediation with no soil relocation. Depending on the chosen upscaling strategy, the depth of soil remediation may vary, ranging from tens of centimeters to several meters. Considering the natural variability of moisture levels in field soil, we assessed the applicability of the HET process for remediating soil with varying moisture content, which shows that the HET process is suitable for soil containing water up to 14% (Supplementary Note 2, Supplementary Fig. 35).

### Life cycle assessment and techno-economic analysis
The energy consumption of the HET process was evaluated. Thanks to its direct heating, ultrafast heating/cooling rate, and rapid treatment capabilities, the HET process is energy-efficient, with an estimated electrical energy consumption of ~420 kWh tonne⁻¹ (Supplementary

Note 3). The energy consumption of the HET process is comparable to, or lower than, that of traditional thermal remediation techniques, including SEE, TCH, ERH, and RFH[59] (Supplementary Table 3, Supplementary Fig. 36). The HET process is also more energy-efficient than other innovative electricity-based remediation techniques, such as the electrochemical method[6] (Supplementary Fig. 36).

A comparative life-cycle assessment (LCA) was conducted to evaluate the environmental impact and energy requirements of the HET process in comparison to established methods for remediating PAH-contaminated soil (Supplementary Note 4, Supplementary Tables 4-5). Four scenarios were examined in this study (Fig. 5a, Supplementary Fig. 37): HET process, thermal desorption[60], soil washing[12], and chemical oxidation[61]. First, as expected, both the HET process and thermal desorption show minimal cumulative water use (CWU), whereas soil washing and chemical oxidation require substantial amounts of water (Supplementary Fig. 38a, Supplementary Table 6). Second, the HET process demonstrates cumulative energy demand (CED) of 3408 MJ tonne⁻¹, which is slightly higher than thermal desorption (2800 MJ tonne⁻¹), but 38–58% lower than soil washing and chemical oxidation (Supplementary Fig. 38b, Supplementary Table 7).

Furthermore, a techno-economic analysis (TEA) was conducted to assess practical applicability (Supplementary Note 4, Supplementary Table 8). Due to its low materials and energy consumption, the HET process has an operating expense of ~$43.3 tonne⁻¹, which is lower than the thermal desorption (~$45.7 tonne⁻¹), soil washing (~$140.2 tonne⁻¹), and chemical oxidation (~$163.0 tonne⁻¹) (Fig. 5b, Supplementary

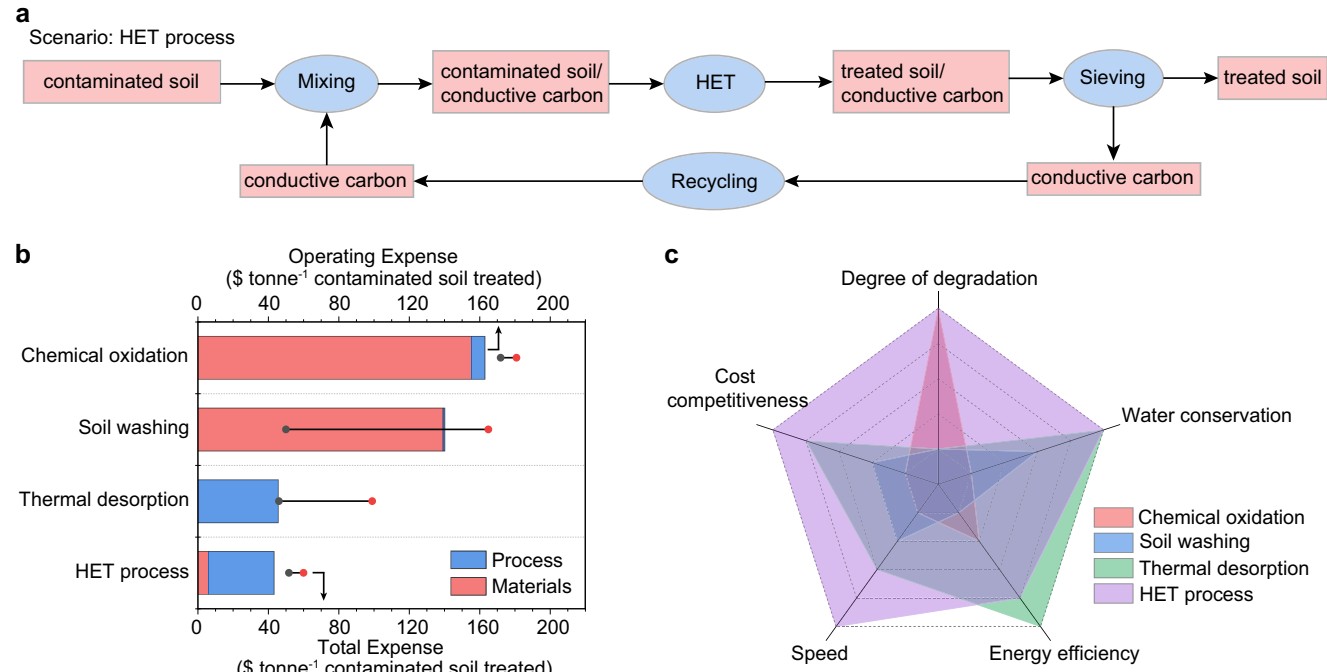

**Fig. 5 | Life-cycle assessment and techno-economic analysis for remediation of PAH-contaminated soil. a** The material flow of the HET process. Other methods (thermal desorption, soil washing, and chemical oxidation) are shown in Supplementary Fig. 37. **b** Economic analysis of various scenarios. The gray dots and red dots denote lower limits and upper limits of total expense, respectively. The range of costs for soil washing and thermal desorption are large when compared to chemical oxidation and the HET process because of the varied methods used in the former processes. **c** Comprehensive comparison of different methods.

Table 9). Even with the inclusion of capital expense, the HET process remained cost-competitive with other established methods (Fig. 5b). With its versatility in remediating soil with multiple pollutants, high degree of pollutant degradation, ultrafast operation speed within seconds to minutes, relatively low energy demand and overall expense, and zero water usage (Fig. 5c), the HET process shows promise for future soil remediation practices, complementing existing thermal desorption and soil washing methods.

## Methods
### Materials
Carbon black (Cabot, Black Pearls 2000, average diameter ~10 nm), metallurgical coke (SunCoke Energy), biochar (Wakefield, Amazon), and plastic pyrolysis ash (Shangqiu Zhongming Eco-Friendly Equipment Co., Ltd in Shangqiu City, Henan, China) was separately used as conductive additive. The Metcoke and Plastic Ash were ground using a mortar and pestle before use. The heavy metal salts used were $HgCl_2$ (99.999%, MilliporeSigma), $Pb(NO_3)_2$ (99.999%, MilliporeSigma), $CoCl_2 \cdot 6H_2O$ (98%, MilliporeSigma), $NiCl_2$ (98%, MilliporeSigma), and $CuCl_2$ (97%, MilliporeSigma). Cd metal (Mallinckrodt Chemical) and HCl (37%, MilliporeSigma) were used to synthesize $CdCl_2$. The Cd metal was dissolved into 1 M HCl, and the $CdCl_2$ was precipitated by evaporating the solution. Clean soil was collected from the Rice University campus, which did not require pre-heating to remove the residual water or moisture. The soil is classified as sandy loam according to the particle size scales. We analyzed the heavy metal content in the clean soil and found it to be well below the safety threshold (Supplementary Fig. 4a). We contaminated the clean soil with the above heavy metal salts at a concentration of Cd (~100 ppm), Hg (~300 ppm), Pb (~1000 ppm), Co (~2000 ppm), Ni (~10,000 ppm), and Cu (~10,000 ppm). The organic contaminants used were pyrene (98%, Acros Organics), fluorene (98%, Acros Organics), and benz[a]anthracene (MilliporeSigma). Conductive additives and the contaminated soil were mixed using a ball miller (MSEsupplies, PMV1-0.4 L) for 30 min at 300 rpm. We

designated the contaminated soil as c-Soil and the remediated soil as r-Soil.

### HET equipment and process
The electrical diagram and picture of the HET system are shown in Supplementary Fig. 2. For the small-scale sample, a mixture of c-Soil (~134 mg) and CB (~66 mg) with a total mass of ~200 mg was loaded into a quartz tube with inner diameter (ID) of 8 mm and outer diameter (OD) of 12 mm. Two graphite electrodes were loosely fit in the quartz tube to permit outgassing and avoid contamination from the metal electrodes during the HET process. The tube was then placed on the reaction stage (Supplementary Fig. 2c) and connected to the HET system (Supplementary Fig. 2b). The reaction stage was put into a desiccator with mild vacuum (~10 mm Hg) to facilitate degassing (Supplementary Fig. 2d). The resistance of the sample was controlled by compressing the electrodes. A capacitor bank with a total capacitance ($C$) of 60 mF was charged by a DC supply, which can reach a voltage up to 400 V. A relay with programmable ms-level delay time was used to control the discharge time. Discharging the capacitor brings the sample to a high temperature. The detailed conditions for the HET process are listed in Supplementary Table 2. After the HET treatment, the apparatus was rapidly cooled to room temperature.

For the enlarged sample, a mixture of c-Soil (~5 g) and Metcoke (~3 g) was loaded into a quartz tube with ID of 1.6 cm and OD of 2.0 cm (Supplementary Fig. 29e). A large-scale HET equipment with $C = 0.624$ F was used (Supplementary Fig. 29a). For the further enlarged sample, a mixture of c-Soil (~100 g) and Metcoke (~30 g) was loaded into a quartz tube with ID of 4.7 cm and OD of 5 cm (Supplementary Fig. 31). An AC source was used to supply the electricity input using an AC-HET system (Supplementary Fig. 30). The concentration of heavy metals or organic contaminants in soil was measured before and after the HET process to determine the removal efficiency of contaminants. **CAUTION:** There is a risk of electrocution if this equipment is used without proper safety constraints. Safety guidelines are listed in the Supplementary Information (Supplementary Fig. 2).

## Characterization

SEM images were obtained using a FEI Quanta 400 ESEM FEG system at 5 kV. XRD patterns were collected using a Rigaku D/Max Ultima II system configured with a Cu Kα radiation ($\lambda = 1.5406$ Å). XPS analyses were conducted using a PHI Quantera XPS system at a base pressure of $5 \times 10^{-9}$ Torr. Elemental spectra were collected with a step size of 0.1 eV with a pass energy of 26 eV. All the XPS spectra were calibrated using the standard C 1 $s$ peak at 284.8 eV. Raman spectra were acquired using a Renishaw Raman microscope (laser wavelength of 532 nm, laser power of 5 mW, 50× lens). The temperature was measured using an infrared (IR) thermometer (Micro-Epsilon) with a temperature range of 1000 to 3000 °C and a time resolution of 1 ms. UV-Vis measurements were carried out on a Shimadzu UV-3600 Plus spectrophotometer. Thermogravimetric analysis (TGA) was conducted in air with a heating rate of 10 °C min$^{-1}$ using the Q-600 Simultaneous TGA/DSC equipment from TA instruments.

XRF was performed using a Panalytical Axios Cement XRF equipment. The test materials (raw soil and soil after HET treatment) were crushed until at least 90% of the material passed a #325 sieve (44 μm). The weight and flux amount of each sample were documented, and the specimens were then prepared into glass beads by fusion using a Katanax K2 Prime. Samples were heated in platinum crucibles to 1000 °C for 15 min while being rocked back and forth for dispersion. Fused lithium metaborate/lithium tetraborate and lithium nitrate were used as fluxing agents. After fusion, the platinum crucibles containing the samples were poured into platinum molds to form beads. The fused beads were then automatically fed into the XRF via the sample loader for continued analysis. The SuperQ analytical software used the documented weights of each sample and its flux weight to generate molar quantitative results.

## Sample digestion and ICP-OES measurement of heavy metals concentration

The standards (Cd, Hg, Pb, Co, Ni, and Cu) were purchased from MilliporeSigma (1000 mg L$^{-1}$ in 2 wt% HNO$_3$). HNO$_3$ (67–70 wt%, TraceMetal$^{TM}$ Grade, Fisher Chemical), HCl (37 wt%, 99.99% trace metals basis, MilliporeSigma), H$_2$O$_2$ (30 wt%, for trace analysis, MilliporeSigma), and ultrapure water (MilliporeSigma Aldrich, ACS reagent for ultratrace analysis) were used for sample digestion. The soil samples (~50 mg) were digested using a modified method from the standard set by the Environmental Protection Agency (EPA), USA[62]. Briefly, the samples (~50 mg) were added to 2 mL of HNO$_3$ (67–70 wt%, 1:1 with water) and heated to 95 °C for 2 h. Then, 2 mL of H$_2$O$_2$ (30 wt%, 1:1 with water) was added and heated to reflux at 95 °C for 2 h. Next, 1 mL of HCl (37 wt%) and 5 mL of H$_2$O were added, and the mixture was heated at reflux for 15 min. The acidic solution was filtered to remove any undissolved particles using a sand core funnel (Class F). The filtrate was then diluted to a range within the calibration curve. ICP-OES measurement was conducted using a Perkin Elmer Optima 8300 ICP-OES system. Prior to measurement, the ICP-OES equipment was carefully calibrated. All samples were measured 3 times to obtain standard deviations. The removal efficiency ($R$) of heavy metals is calculated according to Eq. 1,

$$R = \frac{c(\text{c} - \text{Soil}) \times m(\text{c} - \text{Soil}) - c(\text{r} - \text{Soil}) \times m(\text{r} - \text{Soil})}{c(\text{c} - \text{Soil}) \times m(\text{c} - \text{Soil})} \times 100\% \quad (1)$$

where $m$(c-Soil) is the mass of c-Soil used for HET, $c$(c-Soil) is the concentration of heavy metals in c-Soil, $m$(r-Soil) is the mass of r-Soil after HET, and $c$(r-Soil) is the concentration of heavy metals in r-Soil.

## Solvent extraction of PAH and content determination by UV-Vis spectrophotometry

Calibration curves for pyrene, fluorene, and benz[a]anthracene were prepared by dissolving a known amount of the analyte in ethanol (100%, Decon Laboratories). The solvent extraction method used was modified from the EPA, USA[43]. The extraction solvent consisted of 1:1 $v$:$v$ ethanol:dichloromethane (99.5%, Fischer Chemical). Soil samples (~10 mg) were mixed with the extraction solvent (~5 mL) and dispersed by sonication for 5 min (Cole-Parmer Ultrasonic Cleaner). The resulting solution was filtered using a sand core funnel (Class F) to remove residual soils. The clear filtrate was then diluted with ethanol until the analyte concentration was within the calibration range. Analyte concentration was determined by UV-Vis measurement using a Shimadzu UV-3600 Plus spectrophotometer. This method was validated by spike recovery on each analyte. All the samples were measured three times to obtain the standard deviations.

## GC-MS measurement of PAH

A standard solution of common PAH was purchased from Agilent (Product #G3440-85009), which included 10 μg mL$^{-1}$ of benz[a] anthracene. A serial dilution was performed to calibrate the concentration to the integrated area of the corresponding peak (ranging from 10 ppm to 0.001 ppm), using the same solvent system for analyte sample extraction. The samples were analyzed using an Agilent 8890 GC system, outfitted with a low-bleed J&W HP-5ms capillary column (30 m length, 0.25 mm internal diameter, and 0.25 μm film thickness). The samples were quantified using an Agilent 5977B MSD. A carrier gas of He was used (3.95 psi pressure), with an initial oven temperature of 80 °C held for 2 min, then ramped to 280 °C at a rate of 8 °C min$^{-1}$, and held at 280 °C for 2 min, for a total run time of 29 min. An injection volume of 1 μL was used, with an Agilent 5190-3983 inlet liner. A high (320 °C) inlet temperature was used to facilitate heavy PAH vaporization, and a pulsed splitless injection mode (pulse pressure 50 psi for 0.7 min, followed by purge flow to split vent of 50 psi for 0.75 min to reduce sample-to-sample contamination) to maximize the amount of heavy PAH present in the sample to be transferred onto the column. The transfer line to the MSD was maintained at 320 °C to maximize the analyte signal. To further lower the limit of detection, single ion monitoring at 228 $m$/$z$ was used. Benz[a]anthracene and chrysene have identical $m$/$z$ and fragmentation patterns, and similar retention times, so deuterated chrysene was used to distinguish the peaks. Good linearity in the calibration curve was observed (Supplementary Fig. 19b). Analysis of samples and calibration curve were carried out in triplicate to afford standard deviations.

## Soil particle size distribution measurement

To prepare the samples, we added 1.0 g of raw soil and treated soil into separate 5.0 mL 0.1 M HCl solutions. The carbonate inside the soils was removed by reacting with HCl using an ultrasonic bath (Cole-Parmer Ultrasonic Cleaner) for 15 min. Then, the samples were centrifuged (Adams Analytical Centrifuge, 2 g, 5 min) and washed three times with ultrapure water (MilliporeSigma Aldrich, ACS reagent for ultratrace analysis). Next, 2.0 mL of 35 wt% H$_2$O$_2$ solution was mixed with the soil in a 90 °C water bath for 45 min to remove the organic matter[63]. After another round of centrifugation and water washing three times, the soil particles dispersed in the water were added into the Laser Particle Size Analyzer (ZEN 3600 Zetasizer Nano, Malvern, Worcestershire, UK) for particle size measurement. Based on the measured data, we calculated the ratio of clay (<2 μm), silt (2–50 μm), and sand (>50 μm) in the soil by analyzing the particle size differences.

## Infiltration rate test

The water infiltration rate test was conducted in a laboratory scale using a tube with an inner diameter of 8 mm as the container. A sponge was used to hold the soil samples, allowing for fast penetration of water (Fig. 4d, Supplementary Fig. 25). The water could penetrate the sponge rapidly and did not affect the infiltration test of the soil. Both the raw soil and HET-treated soil, with the same volume, were placed on top of separate sponges. 2 cm of water were then gently added on

top of the soil. The liquid levels were recorded at different times, and the infiltration rate was calculated using the Eq. 2,

$$\text{infiltration rate} = H/t \tag{2}$$

where $H$ is the liquid level in cm and $t$ is the time in h.

## Plant growth

Broccoli sprouts were grown under 16-h light (2.2 mW cm$^{-2}$)/8-h dark cycles at 21 °C in Rice University BRC green house. The raw soil was sterilized by heating it at 120 °C for 2 h. HET-treated soil was mixed with the sterilized raw soil ($w/w$ = 1:1) for plant growth, which is denoted as 50% treated Soil. The sterilized raw soil was used as a control, denoted as raw soil. Before being placed in the container, raw soil or treated soil was mixed with water to obtain the appropriate moisture level. Then, 9 seeds were placed on the wet soil for gemination and growth monitoring. Coffee filters were placed in the soil container to prevent cross-contamination between treated soil and raw soil. The germination rate was calculated by counting the sprouts based on the number of seeds used. We repeated this experiment three times.

## Exchangeable nutrients measurement

The exchangeable P, Ca, K, Mg, Mn, and Fe in raw soil and treated soil were extracted using the Mehlich-3 reagent[64]. The extract is composed of 0.2 M $CH_3COOH$, 0.25 M $NH_4NO_3$, 0.015 M $NH_4F$, 0.013 M $HNO_3$, and 0.001 M ethylenediaminetetraacetic acid (EDTA). 1 g of soil sample was added to 10 g of the extract at a soil-to-solution ratio of 1:10. The mixture was shaken immediately on a horizontal shaker for 5 min. Then, the sample was centrifuged (Adams Analytical Centrifuge, 2 g, 5 min), followed by filtration using a sand core funnel (Class F) to remove any undissolved particles. The filtrate was diluted to the appropriate concentration using 2 wt% $HNO_3$ within the calibration curve range. The P, K, Mg, Mn, and Fe were measured using inductively coupled plasma mass spectrometry (ICP-MS) with a Perkin Elmer Nexion 300 ICP-MS system. Periodic table mix 1 for ICP (10 mg L$^{-1}$, 10 wt% $HNO_3$, Millipore Sigma) was used as the standard for the ICP-MS measurement. Due to interference from Ar, Ca cannot be measured by ICP-MS. Therefore, Ca was measured by ICP-OES using a Perkin Elmer Optima 8300 ICP-OES system. Ca standard (1000 mg L$^{-1}$, 2 wt% $HNO_3$, Millipore Sigma) was used for the ICP-OES measurement.

The soil nitrate-nitrogen serves as an indicator of available nitrogen for plants. The soil nitrate content was measured using ultraviolet second-derivative spectrophotometry to exclude the effect of soluble organic nitrogen in the soil[65]. 1 g of soil sample was added to 10 g of ultrapure water (MilliporeSigma Aldrich, ACS reagent for ultratrace analysis) for the nitrate extraction. The mixture was immersed in an ultrasonic bath (Cole-Parmer Ultrasonic Cleaner) for 15 min. Nitrate standard solutions (0.5 ppm, 1.0 ppm, 2.0 ppm, 4.0 ppm, 10 ppm) were prepared by dissolving an appropriate amount of $NaNO_3$. The UV spectra were acquired using a Shimadzu UV-3600 Plus spectrophotometer (Supplementary Fig. 27a). Then, the second-derivative spectra were computed using Origin 2019 (Supplementary Fig. 27b). The second-derivative absorbances at 223.2 nm were used to determinate the nitrate content. The good linearity of the calibration curve demonstrates the effectiveness of this method (Supplementary Fig. 27c). The nitrate in raw soil and HET treated soil was extracted using DI water. 1 g of soil sample was added to 10 g of DI water, and the mixture was shaken for 5 min. Then, the sample was centrifuged (Adams analytical centrifuge, 60 rpm, 5 min), followed by filtration using a sand core funnel (Class F) to remove any undissolved particles. Finally, the filtrate was diluted to a concentration within the calibration curve range.

## Soil carbon content measurement

The soil carbon content was measured using a ECS 4010 – CHNS-O Elemental Combustion System. Before the measurement, 1.0 g of soil sample was treated with 10 mL of 0.1 M HCl in an ultrasonic bath (Cole-Parmer Ultrasonic Cleaner) for 15 min to remove inorganic carbon (e.g., carbonate). Subsequently, the sample was dried at 105 °C to prepare for the measurement. Acetanilide was used as the standard for calibration. Both the raw soil and the treated soil, after removing carbon additives through sieving, were subjected to carbon content measurement. Analysis of samples was carried out in triplicate to afford standard deviations.

## Life-cycle assessment and techno-economic analysis

The aim of the LCA and TEA is to assess water usage, energy demand, and expense associated with various soil remediation scenarios, including the implementation of the HET process. The scope of the system considered here includes two primary steps: raw materials production and processing. Transportation is not accounted for in this analysis, and it assumes a lab-scale HET process without further scaling. The functional unit used for evaluation is the remediation of 1 tonne of PAH-contaminated soil. The life-cycle inventory is provided in Supplementary Tables 4–9. Energy input of the HET process was measured experimentally, while ISO-compliant GREET database values or data from literatures are employed to calculate CWU, CED, and expenses.

## Data availability

The data supporting the findings of the study are available within the paper and its Supplementary Information. The source data generated in this study have been deposited in the Zenodo database under https://doi.org/10.5281/zenodo.8162061. Source data are provided with this paper.

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

## Acknowledgements
The authors thank Prof. Caroline A. Masiello, Prof. Pedro Alvarez, and Prof. George Hirasaki of Rice University for helpful discussions. The authors thank Dr. Bo Chen of Rice University for helpful input with the XPS results and Dr. Christopher Pennington for developing ICP-OES and ICP-MS methods. The funding of the research is provided by Air Force Office of Scientific Research (FA9550-22-1-0526, J.M.T.), US Army Corps of Engineers, ERDC (W912HZ-21-2-0050, J.M.T., M.G.U.A.) and the Stauffer-Rothwell Scholarship from Rice University (K.M.W). Permission to publish was granted by Director, Geotechnical & Structures Laboratory, ERDC. The characterization equipment used in this project is partly from the Shared Equipment Authority (SEA) at Rice University.

## Author contributions
B.D. and J.M.T. conceived the idea. B.D. conducted the heavy metals removal experiments. R.A.C. and B.D. conducted the PAH removal experiments. D.X.L. and L.E. built the system. L.E. synthesized the flash graphene from Metcoke and maintained the system. M.G.U.A conducted the XRF measurement. Y.L. and J.B. conducted the plant growth. GC-MS measurement was conducted by K.M.W. C.K. built the gas collection apparatus. Y.C. measured the particle size with the help of D.J. and M.A.T. B.D. and Y.C. measured the exchangeable nutrient in soil. B.D., Y.C. and L.E. conducted the scaling up experiment with the help of K.J. and S.X. B.D. and Y.C. measured the soil carbon content with help of X.G. B.D. conducted the LCA and TEA with the help of K.M.W. B.D. and J.M.T. wrote the manuscript. All aspects of the research were overseen by J.M.T. All authors have discussed the results and given approval to the final version of the manuscript.

## Competing interests
An US provisional patent has been filed by Rice University on the HET strategy for soil remediation (J.M.T. and B.D., PCT/US2022/014923), which has not yet been licensed. The authors declare no other competing interests. All other authors declare no competing interests.
