## [Peer Review File · Nature Communications]

REVIEWER COMMENTS

Reviewer #1 (Remarks to the Author):

Comments:

The manuscript titled “High-temperature Electrothermal Remediation of Multi-Pollutant Soils” authored by Bing Deng et al., is interesting and informative. The authors studied the high temperature electrothermal technique to remediate the multiple pollutant in the soil. However, the following are the observations that need to be addressed or clarified by the authors.

Comments:

1.Title: “High-temperature Electrothermal Remediation of Multi-Pollutant Soils”. ‘Soil’ is appropriate for better understanding. If more than one soil is studied, it is better to mention as soils. The title can be “High-temperature Electrothermal Remediation of Multi-Pollutants in Soil”

2.It is mentioned in the manuscript that there is an increase in in water infiltration rate after the HET treatment of soil. But the reason for the increase is not explained clearly.

3.The authors reported that the process is energy-efficient with an electrical energy consumption of ~420 kWh ton⁻¹. Probably, the estimation was done based on the energy consumption for 0.2 g of soil sample (134 mg of soil and 66 mg of carbon black) studied and reported for one tonne. Will the estimation consistent for the scale up of the process in the real field?

4.Line 62: ‘contaminated soils’ can be mentioned as ‘contaminated soil’

5.Lines 62-65: It is mentioned in the statement as “By incorporating carbon conductive additives such as environmentally friendly biochar, the soil temperature can be rapidly increased to 1000 to 3000 °C with a heating rate of ~104 °C s⁻¹ within seconds using pulsed electric input, followed by a rapid cooling rate of ~103 °C s⁻¹”. It is understood that the rapid heating is achieved by using pulsed electric input. How the rapid cooling was achieved? Clarify.

6.Lines 286-288: It is mentioned as “metallurgical coke or plastic pyrolysis ash was used as conductive additive. The Metcoke or Plastic Ash was ground using a mortar and pestle before use”. However, in supplementary material Fig. 7, results of both the materials are reported. Clarify.

7.Lines 296-298: “We contaminated the clean soil with the above heavy metal salts at a concentration of Cd (~100 ppm), Hg (~300 ppm), Pb (~1000 ppm), Co (~2000 ppm), Ni (~10000 ppm), and Cu (~10000 ppm).” Why each metal is used with different concentrations?

8.It is only mentioned that the soil sample was collected from the University campus. What is the classification of soil/ type of soil?

9.Line 427: It is mentioned that the raw soil (sterilized) and HET treated soil was mixed as w/w 1:1 for plant growth (denoted as Treated Soil). How it will be considered as treated soil when raw is mixed w/w 1:1? It is misleading the readers. Clarify.

10.Although the carbon based additives such as carbon black or biochar are added to the soil for ease of conductivity, what is the influence of these materials on infiltration rate, plant growth, exchangeable nutrients?

11.Fig 2.a: What is the reason for reduction of clay and silt, and increase in sand content in the treated soil?

12.Lines 90-92: It is mentioned as “...we also used other carbon sources, such as biochar, which are more environmentally friendly and equally effective”. However, there is nothing reported in this manuscript about experiments on biochar as carbon source. Clarify.

13.Lines 163-164: It is reported as “the heavy metals were detected on the quartz tube side walls”. It raises a concern on transferring the pollutant from one source to the other. Clarify.

14.Although the conceptual designs (HET in tractor, Field assembly, etc.) by the authors are appreciable, the limitations/constraints in the field such as field moisture content, depth of soil to be remediated, adding the carbon additives to the soil in the field, etc are the challenges to have a real scale up work done on the field with a significant soil mass.

15.Line 280: The term “low cost” is not appropriate to use unless the authors really include every cost aspect (not only electricity consumption) of the process (including, cost of carbon additives and any others) and compared with the other technologies. Similarly, the term “good scalability” while working with the sample quantity of 8g (as compared to the huge soil mass for remediation in the real field).

Reviewer #2 (Remarks to the Author):

In this manuscript, Deng et al. reported a high-temperature electrothermal process (HET) process for soil remediation. They used electric pulse to ramp the soil temperature to >1000 °C instantly, under which the heavy metals are removed by evaporation, and the persistent organic pollutants are removed by graphitization. Interestingly, this process leads to the rapid mineralization of soil organic matters, which increases the soil exchangeable nutrients contents and thus improves the soil germination rate. Overall, the application of the high-temperature direct electric heating in soil remediation is novel, compared to other low- to middle-temperature thermal remediation processes. The manuscript is well organized. I think this manuscript is acceptable for publication in Nature Communications after the following issues being addressed:

Major issues:

1. This direct electric heating process is firstly used by the authors in soil remediation in this manuscript. However, this thermal shock method has recently been used in other fields such as materials production (e.g., *Adv. Mater.* 10.1002/adma.202208974 (2022)) and waste management (e.g., *Nano Res.* (2022). <https://doi.org/10.1007/s12274-022-5244-z>). The authors should comprehensively discuss existing pioneering and breakthrough works about the thermal shock process in the Introduction part.
2. The authors proposed the evaporation mechanism for the removal of heavy metals in soil (Fig. 2a). They used metal salts as the contaminates, such as HgCl₂ for Hg, CoCl₂ for Co, NiCl₂ for Ni, etc. As we known, even for a specific element, its different chemical compounds have very different vapor pressures. How does the authors consider the effect of chemical species on the removal of heavy metals? For example, will the removal efficiency the same for HgCl₂ and Hg(0)?
3. For the exchangeable nutrient measurement, the authors observed the increased Fe, P, N, Mn and Ca, and decreased K and Mg after the HET process. Could the authors explain why the K and Mg are reduced but other metals (Fe, Mn, Ca) are increased?
4. Usually the soil has moisture. Does this method only useful for dry soil? Or it can be used for moisture-containing soil? Is pre-drying needed for the use of the HET process for soil remediation?

Other minor issues:

1. The authors provides the temperature profiles of 80 V and 100 V for heavy metal removal. What is the temperature of 60 V and 120 V since you also used these voltages?
2. The authors provides the temperature profile for heavy metal removal. The temperature for remediation of PAHs is also required.
3. The authors used a sieving process for separating carbon additives and soil, and reused the carbon additives. What's the residual carbon content in the soil after the process?

Dear Reviewers:

We greatly appreciate your conscientious reviews on our manuscript entitled “High-temperature electrothermal remediation of multi-pollutants in soil” (NCOMMS-23-18725), which we find to be very beneficial for strengthening our work. We respond to your questions here with point-to-point responses listed below. If you have further questions, we are eager to address those as well. The reviewers’ comments are copied in *italics*, our responses are in bold, and the revision made to the manuscript is marked in **blue and bold**.

Sincerely,

James M. Tour,

On behalf of all authors.

Reviewer #1 (Remarks to the Author):

Comments:

The manuscript titled “High-temperature Electrothermal Remediation of Multi-Pollutant Soils” authored by Bing Deng et al., is interesting and informative. The authors studied the high temperature electrothermal technique to remediate the multiple pollutant in the soil. However, the following are the observations that need to be addressed or clarified by the authors.

Response: We appreciate the referee for their kind review of our work. Point-by-point responses to the concerns are provided below.

Comments:

1.Title: “High-temperature Electrothermal Remediation of Multi-Pollutant Soils”. ‘Soil’ is appropriate for better understanding. If more than one soil is studied, it is better to mention as soils. The title can be “High-temperature Electrothermal Remediation of Multi-Pollutants in Soil”

Response: We agreed with the reviewer. The title has been changed according to the reviewer’s suggestion.

Title:

“High-Temperature Electrothermal Remediation of Multi-Pollutants in Soil”

2.It is mentioned in the manuscript that there is an increase in in water infiltration rate after the HET treatment of soil. But the reason for the increase is not explained clearly.

Response: We thank the reviewer for their comment. Soil texture, or the percentage of sand, silt, and clay in a soil, is the major factor inherent in affecting infiltration (ref: Chapter 5: Soil Water: Characteristics and Behavior, The Nature and Properties of Soil, Edited by Ray R. Weil and Nyle C. Brady, Pearson, 2016, 213-214). Water moves more quickly through the large pores in sandy soil than it does through the small pores in silt and clay soil. According to our soil texture analysis (Fig. 4a, Supplementary Fig. 23), the soil sand ratio is slightly increased after the HET treatment, while the silt and clay contents are slightly decreased. Hence, we think the increased infiltration rate can be ascribed to the increased sand content in the soil. Additionally, the presence of residual carbon in the treated soil can contribute to soil porosity and further facilitate water infiltration.

We revised the manuscript accordingly, on P12,

“The enhanced infiltration rate observed in the HET-treated soil can be ascribed to the increased sand ratio (Fig. 4a, Supplementary Fig. 23), as water tends to flow more rapidly through the large pores in sandy soil compared to the smaller pores in clay soil⁵³. Additionally, the presence of residual carbon in HET-treated soil following the sieving process (Supplementary Fig. 12) could contribute to soil porosity and further facilitate water infiltration.”

Reference:

53. Soil Water: Characteristics and Behavior, The Nature and Properties of Soil, Edited by Weil, R.R. and Brady, N.C., Pearson, 2016, pp. 213-214.

3. The authors reported that the process is energy-efficient with an electrical energy consumption of $\sim 420 \text{ kWh ton}^{-1}$. Probably, the estimation was done based on the energy consumption for 0.2 g of soil sample (134 mg of soil and 66 mg of carbon black) studied and reported for one tonne. Will the estimation consistent for the scale up of the process in the real field?

Response: We thank the reviewer for the comment on energy estimation when scaling up the process. This value of 420 kWh ton^{-1} is based on the small-scale sample (Supplementary Note 1). We here calculated the energy consumption for the upscaled sample. As shown in Supplementary Fig. 29, we scaled up the process to 8 g per batch and obtained the total treated soil of 100 g within 10 min. For the second-generation HET system with $C_2 = 0.624 \text{ F}$ (Supplementary Note 2, Supplementary Fig. 29) and the treatment conditions of $V_1 = 200 \text{ V}$, $V_2 = 0 \text{ V}$, and $M = 8 \text{ g}$, the energy consumption was estimated to be:

$$E = 1.56 \text{ kJ g}^{-1} = 4.37 \times 10^{-4} \text{ kWh g}^{-1} = 437 \text{ kWh ton}^{-1}$$

This matches well the value calculated from the small-scale sample (420 kWh ton⁻¹). Hence, the energy estimation is consistent for scaling up the mass to 40×. Theoretically, the electric energy density plays a crucial role in the HET remediation process. As long as the energy density remains the same during the scaling up process, the estimation of energy consumption will remain consistent.

We have revised the manuscript, on P10 of the SI, Supplementary Note 2,

For the second-generation system with $C_2 = 0.624 \text{ F}$ (Supplementary Note 2, Supplementary Fig. 29), and the upscaled sample parameters $V_1 = 200 \text{ V}$, $V_2 = 0 \text{ V}$, and $M = 8 \text{ g}$, the energy consumption was calculated to be:

$$E = 1.56 \text{ kJ g}^{-1} = 4.37 \times 10^{-4} \text{ kWh g}^{-1} = 437 \text{ kWh ton}^{-1}$$

This value is consistent with the small-scale sample. Theoretically, the electric energy density plays a crucial role in the HET remediation process. As long as the energy density remains the same during the scaling-up process, the estimation of energy consumption will remain consistent.

4.Line 62: ‘contaminated soils’ can be mentioned as ‘contaminated soil’

Response: This has been corrected in the revised manuscript, now on P4 Line 70, “multiple pollutants in **contaminated soil**.”

5.Lines 62-65: It is mentioned in the statement as “By incorporating carbon conductive additives such as environmentally friendly biochar, the soil temperature can be rapidly increased to 1000 to 3000 °C with a heating rate of $\sim 10^4 \text{ °C s}^{-1}$ within seconds using pulsed electric input, followed by a rapid cooling rate of $\sim 10^3 \text{ °C s}^{-1}$ ”. It is understood that the rapid heating is achieved by using pulsed electric input. How the rapid cooling was achieved? Clarify.

Response: We thank the reviewer. The real-time temperature of the soil sample is measured using an IR thermometer, as shown in Fig. 1d. According to the temperature profile, the heating rate is calculated to be $\sim 1.08 \times 10^4 \text{ °C s}^{-1}$, and the cooling rate is calculated to be $\sim 1.88 \times 10^3 \text{ °C s}^{-1}$. The rapid cooling is achieved by efficient heat dissipation resulting from intense thermal radiation of the sample (Figure R1), due to the high radiation emissivity of SiO₂ (the main crystalline material in soil) and carbon (ref: *ACS Nano* 2022, 16, 2577).

We revised the manuscript accordingly, on P5,

“The heating and cooling rates were calculated to be $\sim 1.08 \times 10^4 \text{ }^\circ\text{C s}^{-1}$ and $\sim 1.88 \times 10^3 \text{ }^\circ\text{C s}^{-1}$, respectively. The ultrafast heating is achieved by the pulsed electric input, while the rapid cooling is attributed to the efficient heat dissipation resulting from intense thermal radiation (Supplementary Fig. 3).”

Figure R1. Picture of the soil sample during HET. The rapid cooling is enabled by thermal radiation.

(This figure has been added into the SI as Supplementary Fig. 3)

6.Lines 286-288: It is mentioned as “metallurgical coke or plastic pyrolysis ash was used as conductive additive. The Metcoke or Plastic Ash was ground using a mortar and pestle before use”. However, in supplementary material Fig. 7, results of both the materials are reported. Clarify.

Response: We thank the reviewer. Different carbon sources have different conductivity, cost, and availability. In this work, we used different types of carbon sources as conductive additives, including carbon black, biochar, metallurgical coke, and plastic pyrolysis ash, to demonstrate the wide applicability of carbon sources in our HET process. We here revised the manuscript to use “and” to replace “or”, on P16, “... metallurgical coke, biochar, **and** plastic pyrolysis ash...”, “... Metcoke **and** Plastic Ash...”

7.Lines 296-298: “We contaminated the clean soil with the above heavy metal salts at a

concentration of Cd (~100 ppm), Hg (~300 ppm), Pb (~1000 ppm), Co (~2000 ppm), Ni (~10000 ppm), and Cu (~10000 ppm).” Why each metal is used with different concentrations?

Response: We thank the reviewer for this question. We use different concentrations of the heavy metals because they have different toxicity and safety standards. For example, according to California Office of Environmental Health Hazard Assessment, Human Health Level (HHS�), the safety standards for each are: Cd (1.7 ppm), Hg (18 ppm), Pb (80 ppm), Co (660 ppm), Ni (1600 ppm), and Cu (3000 ppm). We used a lower contamination level for the metals with stricter safety standards. This methodology is used in other soil remediation research, for example, Xu *et al.* contaminated the soil with 10000 ppm Cu, 1000 ppm Pb, and 100 ppm Cd in their work (*Nat. Commun.* 10, 2440 (2019)). In addition, there is large variation in hazardous levels among different contaminated sites, so a wide range of contamination level from 100 – 10000 ppm serves as a good representation for real-world applications.

We have revised the manuscript accordingly, on P6, “**Considering the different toxicity, disparate safety standards, and large variation in hazardous levels among real-world contaminated sites for different heavy metals...**”

8.It is only mentioned that the soil sample was collected from the University campus. What is the classification of soil/ type of soil?

Response: we thank the reviewer for the comment on soil type. The soil can be classified according to the particle size scales (Figure R2). According to our measurement (Fig. 4a, Supplementary Fig. 23), the soil composition is: clay (2.5%), silt (27.8%), and sand (69.7%). Hence, the raw soil is classified as sandy loam. After the HET treatment, the soil composition is clay (1.4%), silt (19.5%), and sand (79.1%). Hence, the treated soil is still classified as sandy loam. The HET remediation process does not change the soil type.

Figure R2. Composition of particle size scales.

We revised the manuscript accordingly, on P11, “Both the raw soil and treated soil were classified as sandy loam”, and on P16, “The soil is classified as sandy loam according to the particle size scales.”

9.Line 427: It is mentioned that the raw soil (sterilized) and HET treated soil was mixed as w/w 1:1 for plant growth (denoted as Treated Soil). How it will be considered as treated soil when raw is mixed w/w 1:1? It is misleading the readers. Clarify.

Response: We thank the reviewer for this comment. By mixing the raw soil with treated soil at 1:1, we were able to leverage both the soil organic matter in the raw soil and the increased exchangeable nutrients in the HET treated soil, allowing for the increased plant growth. We acknowledge that this may lead to potential confusion. To address this concern, we denoted the mixed soil as “50% treated soil” in our revised manuscript, on P12, “denoted as 50% treated soil”, on P22, “which is denoted as 50% treated soil”, and on P37 Caption of Fig. 4g, “50% treated soil”.

10.Although the carbon-based additives such as carbon black or biochar are added to the soil for ease of conductivity, what is the influence of these materials on infiltration rate, plant growth, exchangeable nutrients?

Response: We appreciate the reviewer's question regarding the impact of carbon additives on soil properties. We would like to clarify that in our study, the carbon additives were effectively separated from the soil through a sieving process following the treatment (see discussion on P7 of the manuscript, Supplementary Fig. 10-13). The recovery efficiency of carbon additives was ~93%. It is important to note that all the soil properties were measured on the soil samples after the removal of carbon additives. The residual carbon content in the treated soil was found to be low, ~3.5 wt% based on our measurement (Supplementary Fig. 12). Regarding the effect of this few % carbon content on soil properties, we discuss each aspect individually, as below:

(1) Infiltration rate: The ~3.5 wt% of residual carbon could enhance the infiltration rate. This is due to the lower density of carbon compared to soil minerals, which slightly increases soil porosity, leading to improved infiltration.

(2) Exchangeable nutrients: To measure the content of exchangeable nutrients in the conductive additives, such as carbon black (CB) and Metcoke, we followed the same measurement process as with the soil samples. As shown in Figure R3, the concentrations of various nutrients in CB are: Ca(0.09 ppm), Fe(1.3 ppm), P(0.14 ppm), N(<0.1 ppm), Mn(0.02 ppm), K(69 ppm), and Mg(0.37 ppm). Similarly, in Metcoke, the concentrations are: Ca(0.02 ppm), Fe(85 ppm), P(0.14 ppm), N(<0.1 ppm), Mn(0.02 ppm), K (69 ppm), and Mg(0.37 ppm). Let us consider Metcoke as an example. Taking into account the ~3% carbon residue in the treated soil, the nutrients introduced by the carbon additives are: Ca(0.0006 ppm), Fe(2.55 ppm), P(0.0042 ppm), N(<0.003 ppm), Mn(0.00006 ppm), K(2.1 ppm), and Mg(0.011 ppm). For comparison, the nutrient concentrations in the raw soil are: Ca(4474 ppm), Fe(152 ppm), P(56.7 ppm), N(10.6 ppm), Mn(76.6 ppm), K(245 ppm), and Mg(250 ppm). Therefore, the measurement errors introduced by the residual carbon are: Ca($1.3 \times 10^{-5}\%$), Fe(1.7%), P($7.4 \times 10^{-3}\%$), N(<0.028%), Mn($7.8 \times 10^{-5}\%$), K(0.86%), and Mg($4.4 \times 10^{-3}\%$). In conclusion, the carbon additives have little effect on the exchangeable nutrient concentrations.

(3) Plant Growth: The growth of plants is primarily influenced by soil nutrients. Since the 3 wt% carbon additives have little effect on the concentrations of exchangeable nutrients, as demonstrated above, the residual carbon additives have no substantial impact on plant growth.

Figure R3. Exchangeable nutrients measurement in carbon additives, including CB and Metcoke. Note that N content is lower than 0.1 ppm (the detection limit of UV-Vis spectrometry).

(This figure has been added into SI as Supplementary Fig. 28)

We have revised the manuscript accordingly, on P12, “**Additionally, the presence of residual carbon in HET-treated soil following the sieving process (Supplementary Fig. 12) might contribute to soil porosity and further facilitate water infiltration.**”, and on P12, “**Note that the use of carbon conductive additives does not introduce significant error in the measurement (Supplementary Fig. 28).**”

11.Fig. 4a: What is the reason for reduction of clay and silt, and increase in sand content in the treated soil?

Response: We thank the reviewer. The particle size ranges for each are: clay (<2 μm), silt (2 to 50 μm), and sand (>50 μm). The raw soil composition is clay (2.5%), silt (27.8%), and sand (69.7%). After the HET treatment, the soil composition is clay (1.4%), silt (19.5%), and sand (79.1%). The clay and silt content are slightly reduced, while sand content is increased. The reason is that the HET is a high-temperature thermal process, leading to the soil aggregation by fusion of the smaller particles.

We have revised the manuscript accordingly, on P11, “**The treated soil exhibits a slightly**

higher sand composition compared to the raw soil, primarily attributed to soil aggregation during the HET process”.

12.Lines 90-92: It is mentioned as “....we also used other carbon sources, such as biochar, which are more environmentally friendly and equally effective”. However, there is nothing reported in this manuscript about experiments on biochar as carbon source. Clarify.

Response: We appreciate the reviewer for this comment on the use of biochar. In the first version of the manuscript, we already demonstrated the applicability of some carbon sources as conductive additives, including carbon black, metallurgical coke, flash graphene, and plastic pyrolysis ash. Biochar with sufficient conductivity could also be used as the conductive additive. We here use the remediation of pyrene-contaminated soil as an example to show the applicability of biochar. Biochar is mainly composed of carbon with some inorganic constituents (Figure R4a-b). Like other conductive additives, we mixed the contaminated soil with biochar with a mass ratio of 2:1. The HET treatment conditions remained the same as those with carbon black additives (Supplementary Table 2). As shown in Figure R4c, after the HET treatment, the pyrene content in soil is greatly reduced. The removal efficiency of pyrene by just one HET pulse reaches up to ~95% (Figure R4d), which is comparable to the performance of other carbon additives like carbon black (Fig. 3).

Figure R4. Remediation of pyrene-contaminated soil using biochar as conductive additive. (a) XRD pattern of biochar. (b) Raman spectrum of biochar. (c) UV-Vis absorption spectra of extracts from pyrene-contaminated soil and HET treated soil. (d) Removal efficiencies of pyrene.

(This figure has been added into the SI as Supplementary Fig. 20)

We have revised the manuscript accordingly, on P10, **“In addition to carbon black, other environmentally friendly carbon sources like biochar could be used as the conductive additives and shows similar effectiveness (Supplementary Fig. 20).”**

13.Lines 163-164: It is reported as “the heavy metals were detected on the quartz tube side walls”. It raises a concern on transferring the pollutant from one source to the other. Clarify.

Response: We appreciated the reviewer’s question. As shown in Supplementary Fig. 15, we conducted a mass balance analysis of the heavy metals during the HET process. In a typical HET process, the ratios of heavy metals on the quartz tube are Cd (~0%), Pb (~25%), Co (~2%), Ni (~5%), Hg (~5%), and Cu (~9%). It is important to note that the quartz tube can be reused, and the heavy metals deposited on the quartz tube can be effectively collected and prevented from being released into the environment. Therefore, the HET process does not involve the transfer of pollutants; rather, it enables the collection of pollutants by capturing them on the quartz tube or the vacuum trap. The collected heavy metals can then be properly handled to ensure minimal environmental impact.

We have revised the manuscript to clarify this, on P8, **“The evaporated or deposited heavy metals can be captured and properly handled, preventing them from being released into the environment.”**

14. Although the conceptual designs (HET in tractor, Field assembly, etc.) by the authors are appreciable, the limitations/constraints in the field such as field moisture content, depth of soil to be remediated, adding the carbon additives to the soil in the field, etc. are the challenges to have a real scale up work done on the field with a significant soil mass.

Response: We appreciate the reviewer’s concerns regarding the challenges of scaling up the HET process in real field conditions. While we acknowledge these challenges, we would like

to emphasize that our current manuscript serves as proof-of-concept study conducted at the bench scale. However, we have taken steps to address the scaling-up concerns raised by the reviewer, and we propose strategies to overcome these challenges.

(1) Scaling up the sample mass to 100 g per batch.

In our manuscript, we demonstrated the treatment of soil with a mass volume of ~8 g per batch and ~100 g within 10 minutes using the second-generation DC-HET system (Supplementary Fig. 29). To address the scalability concern, we integrated an alternating current (AC) supply with the HET process, which offers better scalability compared to the direct current (DC) process. We have extended the AC supply to treat a sample mass of 100 g per batch (Figure R5). Pyrene contaminated soil was used as a representative. The soil was mixed with Metcoke as the conductive additive and loaded into a quartz tube with an inner diameter of 4 cm (Figure R5a). The sample was connected to the AC system for thermal treatment (Figure R5b). After treatment, a mixture of treated soil and Metcoke was obtained (Figure R5c), and the Metcoke was removed by sieving, resulting in separated treated soil (Figure R5d). After two cycles of AC-HET treatment, the pyrene concentration in the soil was reduced to below the safe level (Figure R5e-f), demonstrating the effectiveness of the HET process for the removal of PAH. The treatment process took ~1 min for the 100 g soil treatment. This production rate corresponds to approximately 6 kg/h or 144 kg/day, which we believe is sufficient to demonstrate feasibility at the laboratory scale.

Nonetheless, we acknowledge the reviewer's comment and agree, there will be much engineering between this lab-scale demonstration and the real field work. We met with the inventors of the Shell process that is now used worldwide for soil remediation and oil contamination. Operationally, they use large heating coils inserted into the ground, with gas capture blankets and vacuum on the surface. So, some of the requirements are similar. Through our process is much faster and lower overall energy, as seen in the LCA. In the end, we concur – there will be much to do in any transition.

Figure R5. Scaling up sample mass to 100 g per batch using AC system. (a) Picture of the 100 g soil sample mixed with Metcoke as conductive additives. (b) Picture of the sample for HET treatment. (c) Picture of the mixture of soil/Metcoke after the HET treatment. (e) Separated soil after sieving process. (f) UV-Vis absorption spectra of extracts from pyrene-contaminated soil before and after repetitive HET treatment. (g) The content of pyrene in soil varied with repetitive HET treatment.

(This figure has been added into the SI as Supplementary Fig. 31)

The above discussion has been added into the SI, on P7 under Supplementary Note 2, “**2.3 Use of the AC-HET process for scaling up to 100 g per batch.**”

(2) Field moisture content.

We have also tested the applicability of the HET process for remediation of moisture-containing soil. We measured the moisture content of the soil used in our experiments. The previously used dry soil had a moisture content of ~1.2% (Figure R6a). We collected another batch of soil with a moisture content of ~14% (denoted as Moisture soil, Figure R6b). Pyrene, a representative contaminant, was added to the moisture soil, which was then mixed with

carbon black as the conductive additive. The electric input of the HET treatment was the same as that for the dry soil (Supplementary Table 2). The intensity of pyrene absorption peaks progressively decreased with increasing electric pulses (Figure R6c-d), like the results obtained from the dry soil (Figs. 3a-b). The pyrene removal efficiency in the moisture soil reaches 91% after 3 HET pulses, which is slightly lower than that of the dry soil (95%). This demonstrates the feasibility of the HET process for remediation of moisture-containing soil.

Figure R6. Remediation of moist soil. (a) TGA curve of dry soil. (b) TGA curve of moist soil. TGA was conducted in air under 110 °C, with a heating rate of 25 °C min⁻¹, for 30 min. (c) UV-Vis absorption spectra of extracts from pyrene-contaminated moisture soil before and after repetitive HET pulses. (d) The content of pyrene in soil varied with repetitive HET pulses. (e) Removal efficiency of pyrene in dry soil varied with repetitive HET pulses. (f) Removal efficiency of pyrene in moist soil varied with repetitive HET pulses.

(This figure has been added into SI as Supplementary Fig. 35)

The above discussion has been added into the SI, on P9 under Supplementary Note 2, “**2.6 Consideration of the soil moisture.**”

(3) The depth of soil.

We have presented three strategies for the field application of the HET process. In the first strategy (*ex-situ* continuous processing, Supplementary Fig. 32), the soil can be excavated from the contaminated site. The depth of soil is not a constraint since excavators can remove soil to the desired depth. In the second strategy (*on-site* tractor processing, Supplementary Fig. 33), the depth of soil will be determined by the disc ploughing machine. Typically, the penetration depth of a disc plough ranges from 3 to 15 inches (7.6 to 38 cm) (Source: <http://ecoursesonline.iasri.res.in/mod/page/view.php?id=482>). In the third strategy (*on-site* vacuum well facility, Fig. 1a, Supplementary Fig. 34), the depth is determined by the electrodes inserted into the soil and the vacuum well. This facility is adapted from traditional *in situ* thermal desorption technologies, which allow for the remediation of soil depths of 4.5 to 6.5 m, as mentioned in the reference (M. Makoudi, A. Jordens, and J. Haemers, *In Situ Thermal Desorption using Smart Burners technology in urban area for the treatment of -VOCl- contaminated soil, 2021, HAEMERS Technologies*). Therefore, the depth of soil remediation can vary depending on the chosen upscaling strategy, ranging from tens of centimeters to several meters.

(4) The addition of carbon additives.

On the bench scale, the mixing of carbon additive with soil can be easily accomplished using a hand mixer, ball miller, and or other mixing machines. For scaling up, the addition of carbon additives depends on the chosen upscaling strategy. In the *ex-situ* continuous processing strategy, the mixing of soil with carbon additive can be realized using a commercial mixer, as illustrated in Supplementary Fig. 33. In the *on-site* tractor processing strategy, the addition of carbon additive can be done by a small-scale mixer, as illustrated in Supplementary Fig. 34. In the *on-site* vacuum well facility strategy, the addition of carbon additives can be combined with the disc ploughing process. After disc ploughing the soil, carbon additives can be applied to the soil surface, followed by another round of disc ploughing for through mixing.

In conclusion, we agree that the field application of the HET process requires careful consideration of the scaling-up equipment and facilities, soil moisture, depth of soil, the

addition of carbon additives, and so on. We have proposed strategies to address these challenges, which are dependent on the chosen upscaling approach. However, it is important to note that our current manuscript represents a proof-of-concept study conducted at the bench scale. Future research and development are indeed necessary to further explore the real-world application of this method and tackle the challenges associated with scaling up. We appreciate the reviewer's valuable feedback, which allows us to emphasize the need for continued research in this direction.

We have revised the manuscript accordingly:

The discussion on upscaling has been included under the subtitle on P13, “**Upscaling strategies and field application potential**”

On P13, “**The AC-HET process is more suitable for scaling up. By using the AC-HET, we realized the remediation of pyrene-contaminated soil with mass of 100 g per batch, with a retention time of ~1 min, resulting in a potential production rate of >100 kg day⁻¹ in a laboratory system (Supplementary Fig. 31, Supplementary Note 2). The upscaled sample exhibited comparable pyrene removal efficiency to that of small-scale samples.**”

On P14, “**Depending on the chosen upscaling strategy, the depth of soil remediation may vary ranging from tens of centimeters to several meters.**”

On P14, “**Considering the natural variability of moisture levels in field soil, we assessed the applicability of the HET process for remediating soil with varying moisture content, which shows that the HET process is suitable for soil containing moisture up to 14% (Supplementary Note 2, Supplementary Fig. 35).**”

On P48 of SI following Supplementary Fig. 32, “**The soil can be excavated from the contaminated sites using excavators. The soil and carbon additives can be mixed using a commercial mixer.**”

On P49 of SI following Supplementary Fig. 33, “**The depth of soil is determined by the disc ploughing machine with typical penetration depth ranging from several to tens of centimeters.**” “**Adding carbon additives and mixing them with the dried soil using a mixer.**”

On P50 of SI following Supplementary Fig. 34, “**The depth of soil is determined by the electrodes inserted into the soil and the vacuum well. This facility is adapted from traditional in-situ thermal desorption technologies, which allow for the remediation of soil depths ranging from 4.5 to 6.5 m, as mentioned in the reference²⁹.**”

15.Line 280: The term “low cost” is not appropriate to use unless the authors really include every

cost aspect (not only electricity consumption) of the process (including, cost of carbon additives and any others) and compared with the other technologies. Similarly, the term “good scalability” while working with the sample quantity of 8g (as compared to the huge soil mass for remediation in the real field).

Response: We appreciate the reviewer’s comments regarding cost and scalability of the process. We would like to address these points as follows:

(1) Scalability: As mentioned in our previous response to comment #14, we have scaled up the process to a sample mass of 100 g per batch within 1 min, equivalent to $>100 \text{ kg day}^{-1}$. In a lab system. We also proposed various solutions to address the scaling concerns raised by the reviewer, including soil moisture content, soil depth, and the addition of carbon additives. However, we agree with the reviewer that the term "good scalability" may be slightly overstated, and we have revised it to "**potential scalability**" in the revised version. It is important to note that our current study serves as a proof-of-concept at the bench scale, and further upscaling work is necessary in the future.

(2) Cost: We have conducted a comprehensive life-cycle assessment (LCA) and techno-economic analysis of the HET process, incorporating all materials, energy, process, equipment cost, and compared it with existing methods such as thermal desorption, soil washing, and chemical oxidation. Four scenarios were examined in this study, as shown in Figure R7, with detailed description of each scenario, system boundaries, life-cycle inventory, life-cycle impact assessment, cost evaluation, and uncertainty considerations provided in Supplementary Note 4.

Summarizing the results, the life-cycle assessment reveals two key conclusions: First, both the HET process and thermal desorption show minimal cumulative water use (CWU), whereas soil washing, and chemical oxidation require substantial amounts of water (Figure R8a, Supplementary Table 6). Second, the HET process demonstrates cumulative energy demand (CED) of 3408 MJ ton^{-1} , which is slightly higher than thermal desorption (2800 MJ ton^{-1}), and 38-58% lower than soil washing and chemical oxidation (Figure R8b, Supplementary Table 7).

Additionally, a techno-economic analysis (TEA) was conducted to assess practical applicability (Supplementary Note 4, Supplementary Table 8). Due to its low materials and energy consumption, the HET process has an operating expense (the addition of materials and energy expense) of $\sim\$43.3 \text{ ton}^{-1}$, which is lower than the thermal desorption ($\sim\$45.7 \text{ ton}^{-1}$).

¹), soil washing (~\$140.2 ton⁻¹), and chemical oxidation (~\$163.0 ton⁻¹) (Figure R9b, Supplementary Table 9). Even with the inclusion of capital expense, the HET process remained cost-competitive with other established methods (Figure R9b).

We have also provided a comprehensive comparison of the four methods in terms of degree of degradation of contaminants, cumulative water use, cumulative energy demand, speed, and total expense, as shown in Figure R9c. The HET process can operate within seconds or minutes, significantly faster than the other methods. The HET process degrades the PAH, while the soil washing, and thermal desorption removes the contaminants by physical separation. With its versatility in remediating multiple pollutants, ultrafast operation speed, relatively low energy demand and overall expense, and zero water usage, we think that the HET process holds promise as a potential approach for future soil remediation, complementing existing thermal desorption and soil washing methods.

Scenario 1: HET process

Scenario 2: thermal desorption

Scenario 3: soil washing

Scenario 4: chemical oxidation

Figure R7. Scenarios for the life-cycle assessment and techno-economic analysis. Materials flow for HET process, thermal desorption, soil washing, and chemical oxidation.

(This figure has been added into SI as Supplementary Fig. 37)

Figure R8. Life-cycle impact assessment. (a) Cumulative water use of various scenarios. (b) Cumulative energy demand of various scenarios.

(This figure has been added into SI as Supplementary Fig. 38)

Figure R9. Life-cycle assessment and techno-economic analysis for remediation of PAH-contaminated soil. (a) The material flow of the HET process. Other methods (thermal desorption, soil washing, and chemical oxidation) are shown in Supplementary Fig. 37. (b) Economic analysis of various scenarios. The range of costs for soil washing and thermal desorption are large when compared to chemical oxidation and the HET process because of the varied methods used in the former processes. (c) Comprehensive comparison of different methods.

(This figure has been added into the manuscript as Fig. 5)

We have revised the manuscript accordingly:

On P2 Abstract, “We propose strategies for *ex-situ* upscaling and field applications. Techno-economic analysis indicates the process holds the potential for being more energy-efficient and cost-effective compared to soil washing or thermal desorption.”

On P15, “A comparative life-cycle assessment (LCA) was conducted to evaluate the environmental impact and energy requirements of the HET process in comparison to established methods for remediating PAH-contaminated soil (Supplementary Note 4, Supplementary Tables 4-5). Four scenarios were examined in this study (Fig. 5a, Supplementary Fig. 37): HET process, thermal desorption, soil washing, and chemical oxidation. First, as expected, both the HET process and thermal desorption show minimal cumulative water use (CWU), whereas soil washing, and chemical oxidation require substantial amounts of water (Supplementary Fig. 38a, Supplementary Table 6). Second, the HET process demonstrates cumulative energy demand (CED) of 3408 MJ ton⁻¹, which is slightly higher than thermal desorption (2800 MJ ton⁻¹), and 38-58% lower than soil washing and chemical oxidation (Supplementary Fig. 38b, Supplementary Table 7).

Furthermore, a techno-economic analysis (TEA) was conducted to assess practical applicability (Supplementary Note 4, Supplementary Table 8). Due to its low materials and energy consumption, the HET process has an operating expense of ~\$43.3 ton⁻¹, lower than the thermal desorption (~\$45.7 ton⁻¹), soil washing (~\$140.2 ton⁻¹), and chemical oxidation (~\$163.0 ton⁻¹, Fig. 5b, Supplementary Table 9). Even with the inclusion of capital expense, the HET process remained cost-competitive with other established methods (Fig. 5b). With its versatility in remediating multiple pollutants, ultrafast operation within seconds to minutes, relatively low energy demand and overall expense, and zero water usage (Fig. 5c), the HET process shows promise for future soil remediation practices, complementing existing thermal desorption and soil washing methods.”

On P24 Method section,

“Life-cycle assessment and techno-economic analysis.

The aim of the LCA and TEA is to assess water usage, energy demand, and expense associated with various soil remediation scenarios, including the implementation of the HET process. The scope of the system considered here includes two primary steps: raw materials production and processing. Transportation is not accounted for in this analysis, and it assumes a lab-scale HET without further scaling. The functional unit used for evaluation is the remediation of 1 ton of PAH-contaminated soil. The life-cycle inventory is provided in Supplementary Tables 4-9. Energy input of the HET process was measured experimentally,

while ISO-compliant GREET database values or data from literatures are employed to calculate CWU, CED, and expenses.”

Reviewer #2 (Remarks to the Author):

In this manuscript, Deng et al. reported a high-temperature electrothermal process (HET) process for soil remediation. They used electric pulse to ramp the soil temperature to >1000 °C instantly, under which the heavy metals are removed by evaporation, and the persistent organic pollutants are removed by graphitization. Interestingly, this process leads to the rapid mineralization of soil organic matters, which increases the soil exchangeable nutrients contents and thus improves the soil germination rate. Overall, the application of the high-temperature direct electric heating in soil remediation is novel, compared to other low- to middle-temperature thermal remediation processes. The manuscript is well organized. I think this manuscript is acceptable for publication in Nature Communications after the following issues being addressed:

Response: We appreciate the reviewer for the positive evaluation to our work. Point-by-point response to the reviewers' comments are shown below.

Major issues:

1. This direct electric heating process is firstly used by the authors in soil remediation in this manuscript. However, this thermal shock method has recently been used in other fields such as materials production (e.g., Adv. Mater. 10.1002/adma.202208974 (2022)) and waste management (e.g., Nano Res. (2022). <https://doi.org/10.1007/s12274-022-5244-z>). The authors should comprehensively discuss existing pioneering and breakthrough works about the thermal shock process in the Introduction part.

Response: We thank the reviewer for the comment about the research background of this method. As pointed out by the reviewer, the direct electric heating is first used by us in soil remediation. It has recently been used in other fields including materials production and processing, waste management and upcycling. We revised the Introduction part of this manuscript to discuss the previous works and supplemented some relevant references, on P3, “Recently, electric heating has emerged as a rapid, energy-efficient thermal treatment process for materials production²² and waste management²³. By designing the direct Joule

heating process, metal nanoparticles²⁴ and high-entropy alloy nanoparticles²⁵ were synthesized through thermal shock, which has found widespread application in the production of functional materials for energy storage^{26,27} and catalysis^{28,29}. Our group developed the flash Joule heating method for converting carbon resources into graphene materials³⁰. Furthermore, the flash Joule heating process has been extended to include waste management applications such as plastic upcycling³¹, critical metals recovery³²⁻³⁴, and battery recycling^{35,36}.

References

- 22 Jiang, R. *et al.* Ultrafast synthesis for functional nanomaterials. *Cell Rep. Phys. Sci.* 2, 100302 (2021).
- 23 Wyss, K. M., Deng, B. & Tour, J. M. Upcycling and urban mining for nanomaterial synthesis. *Nano Today* 49, 101781 (2023).
- 24 Chen, Y. *et al.* Ultra-fast self-assembly and stabilization of reactive nanoparticles in reduced graphene oxide films. *Nat. Commun.* 7, 12332 (2016).
- 25 Yao, Y. *et al.* Carbothermal shock synthesis of high-entropy-alloy nanoparticles. *Science* 359, 1489-1494 (2018).
- 26 Zhu, W. *et al.* Ultrafast non-equilibrium synthesis of cathode materials for Li-ion batteries. *Adv. Mater.* 35, 2208974 (2023).
- 27 Zhang, J. *et al.* Ultrafast manufacturing of ultrafine structure to achieve an energy density of over 120 Wh kg⁻¹ in supercapacitors. *Adv. Energy Mater.* 13, 2203061 (2023).
- 28 Liu, C. *et al.* Multiple twin boundary-regulated metastable Pd for ethanol oxidation reaction. *Adv. Energy Mater.* 12, 2103505 (2022).
- 29 Liu, S. *et al.* Extreme environmental thermal shock induced dislocation-rich Pt nanoparticles boosting hydrogen evolution reaction. *Adv. Mater.* 34, 2106973 (2022).
- 30 Luong, D. X. *et al.* Gram-scale bottom-up flash graphene synthesis. *Nature* 577, 647-651 (2020).
- 31 Wyss, K. M. *et al.* Upcycling of waste plastic into hybrid carbon nanomaterials. *Adv. Mater.* 35, 2209621 (2023).
- 32 Deng, B. *et al.* Urban mining by flash Joule heating. *Nat. Commun.* 12, 5794 (2021).
- 33 Deng, B. *et al.* Rare earth elements from waste. *Sci. Adv.* 8, eabm3132 (2022).
- 34 Deng, B. *et al.* Heavy metal removal from coal fly ash for low carbon footprint cement. *Commun. Eng.* 2, 13 (2023).
- 35 Luo, J. *et al.* Recycle spent graphite to defect-engineered, high-power graphite anode.

Nano Res. 16, 4240-4245 (2023).

36 Chen, W. *et al.* Flash recycling of graphite anodes. *Adv. Mater.* 35, 2207303 (2023).

2. The authors proposed the evaporation mechanism for the removal of heavy metals in soil (Fig. 2a). They used metal salts as the contaminates, such as HgCl₂ for Hg, CoCl₂ for Co, NiCl₂ for Ni, etc. As we known, even for a specific element, its different chemical compounds have very different vapor pressures. How does the authors consider the effect of chemical species on the removal of heavy metals? For example, will the removal efficiency the same for HgCl₂ and Hg(0)?

Response: We appreciate the reviewer's comment on the impact of speciation on heavy metal removal efficiencies. We agree with the reviewer that different chemical compounds have different vapor pressures, which affect the removal efficiency of heavy metals by the HET process. We here considered the chemical species on the removal of heavy metals.

In the HET process, the elevated temperatures can initiate various reactions, including carbothermic reduction facilitated by the presence of carbon conductive additives, thermal decomposition, and evaporation. Consequently, there are multiple potential pathways for the removal of different heavy metals species: (1) Direct evaporation of metal species; (2) Thermal decomposition of metal species to other species followed by their evaporation; and (3) Carbothermic reduction of metal species to lower-valence-state species followed by their evaporation. Given that the temperature in the HET process can reach up to 3000 °C, all the above reactions have the potential to occur, providing various pathways for the removal of different heavy metal species.

In this study, per the reviewer's suggestion, we focused on Hg as an example to investigate the impact of speciation on removal efficiency. The Hg contaminants present in the soil primarily exist in the form of Hg(0) and Hg(II). Depending on the counterions, Hg(II) species can include HgS, HgO, HgCl₂, HgSO₄, and so on¹. To understand the potential reactions involved, we conducted a thermodynamic analysis using the HSC Chemistry 10 software.

For Hg(0), direct evaporation occurs as represented by:

For HgCl₂, it can undergo direct evaporation:

Alternatively, HgCl₂ can decompose to Hg followed by evaporation:

Regarding HgO, direct evaporation can take place:

Alternatively, HgO can thermally decompose to Hg followed by the evaporation of Hg:

Furthermore, HgO may also undergo carbothermic reduction to Hg followed by the evaporation of Hg:

For HgSO₄, direct evaporation can occur:

HgSO₄ can also thermally decompose to Hg followed by the evaporation of Hg:

Based on our thermodynamic analysis (Figure R10a), the decomposition and carbothermic reduction reactions are found to be spontaneous at temperatures >1200 °C, which can be achieved using our HET process. Additionally, the vapor pressure of Hg and HgCl₂ is high at temperatures <500 °C (Figure R10b). Collectively, this analysis suggests that the HET process is viable for removal of Hg with different species.

Next, we used the HET process to remove Hg, HgO, and HgSO₄ from contaminated soil. These individual Hg species were added separately to the soil, which was then mixed with carbon black as the conductive additive. The HET conditions remained consistent (100 V, 1 s) for all Hg contaminants (Figure R10c). Subsequently, we measured the removal efficiencies of each Hg species. As shown in Figure R10d, the HET process achieved high removal efficiencies for all Hg species: Hg (~90.4%), HgCl₂ (~94.6%), HgO (~95.1%), and HgSO₄ (~86.5%) using a single HET pulse of 1 s. Note that HgCl₂ possesses a higher vapor pressure compared to Hg (Figure R10b), leading to its slightly higher removal efficiency. Additionally, HgO is prone to decomposition into Hg (Figure R10a), thereby exhibiting a high removal efficiency. Conversely, the decomposition of HgSO₄ is relatively more challenging and occurs at higher temperatures (Figure R10a), resulting in a slightly lower removal efficiency compared to the other Hg species. Finally, it might be routine to provide more than one pulse, so this is a lower bound.

The above discussion has been added into SI on P2 as “Supplementary Note 1. Influence of chemical species on removal efficiencies of heavy metals.”

Figure R10. Thermodynamic analysis of Hg species. (a) Gibbs free energy changes of the reactions. (b) Vapor pressure-temperature relationship for Hg species. (c) Current input for the remediation of Hg-contaminated soil. (d) Removal efficiencies of different Hg species. (This figure has been added into SI as Supplementary Fig. 16)

We made revision to the manuscript accordingly, on P9, “Contaminated soil containing heavy metals exhibits a wide range of speciation. We analyzed the influence of chemical species on the removal efficiency (Supplementary Note 1). Depending on the thermal properties of the heavy metal species, the elevated temperatures in the HET process can initiate a sequence of reactions, including evaporation, thermal decomposition, and carbothermic reduction. We here considered Hg as an example. Under the HET process (Supplementary Figs. 16a-b), representative Hg compounds such as HgCl_2 , HgO , and HgSO_4 can be converted to Hg at temperatures below 1200 °C. By using a single HET pulse, high removal efficiencies were achieved for all tested Hg species (Supplementary Figs. 16c-d): Hg (~90.4%), HgCl_2 (~94.6%), HgO (~95.1%), and HgSO_4 (~86.5%). This further demonstrates the broad applicability of the HET method for remediating heavy metal.”

3. For the exchangeable nutrient measurement, the authors observed the increased Fe, P, N, Mn

and Ca, and decreased K and Mg after the HET process. Could the authors explain why the K and Mg are reduced but other metals (Fe, Mn, Ca) are increased?

Response: We appreciate the reviewer's comment on exchangeable nutrients change. We observed an increase in exchangeable Fe, P, N, Mn, and Ca, while exchangeable K and Mg showed a slightly reduction (Fig. 4i). The dominant increase in exchangeable nutrients can be attributed to the mineralization of soil organic matter during the HET process. Conversely, the high temperature process can lead to evaporative loss of metals. As a rough estimation, we compared the boiling points of different metals. K (b.p. = 759 °C) and Mg (b.p. = 1090 °C) have much lower boiling points than the other metals including Fe (b.p. = 2861 °C), Mn (b.p. = 2061 °C), and Ca (b.p. = 1484 °C). As shown in Supplementary Fig. 17, the temperature for PAH remediation is ~1500 °C. Under this temperature, K and Mg could experience some degree of loss, leading to a reduction in their exchangeable content.

We revised the manuscript accordingly, on P12, **“The slight decrease in exchangeable K and Mg may be attributed to their higher volatility, leading to the evaporative losses, when compared to other metals like Ca, Fe, and Mn.”**

4. Usually the soil has moisture. Does this method only useful for dry soil? Or it can be used for moisture-containing soil? Is pre-drying needed for the use of the HET process for soil remediation?

Response: We appreciate the reviewer's comment on the impact of soil moisture. This same concern was raised by the first reviewer shown in comment #14.

We have tested the applicability of the HET process for remediation of moisture-containing soil. We measured the moisture content of the soil used in our experiments. The previously used dry soil had a moisture content of ~1.2% (Figure R6a). We collected another batch of soil with a moisture content of ~14% (denoted as Moist soil, Figure R6b). Pyrene, a representative contaminant, was added to the moisture soil, which was then mixed with carbon black as the conductive additive. The electric input of the HET treatment was the same as that for the dry soil (Supplementary Table 2). The intensity of pyrene absorption peaks progressively decreased with increasing electric pulses (Figure R6c-d), like the results obtained from the dry soil (Figs. 3a-b). The pyrene removal efficiency in the moisture soil reaches 91% after 3 HET pulses, which is slightly lower than that of the dry soil (95%). This demonstrates the feasibility of the HET process for remediation of moisture-containing soil

up to 14%. Pre-drying is not required for such moisture level. However, if the soil is very wet (for example, > 30%), pre-drying would be required.

Figure R6. Remediation of moist soil. (a) TGA curve of dry soil. (b) TGA curve of moist soil. TGA is conducted in air with a heating rate of 25 °C, and then kept at 110 °C for 30 min. (c) UV-Vis absorption spectra of extracts from pyrene-contaminated moist soil before and after repetitive HET pulses. (d) The content of pyrene in soil varied with repetitive HET pulses. (e) Removal efficiency of pyrene in dry soil varied with repetitive HET pulses. (f) Removal efficiency of pyrene in moist soil varied with repetitive HET pulses.

(This figure has been added into SI as Supplementary Fig. 35)

We have revised the manuscript accordingly, on P14, “Considering the natural variability of moisture levels in field soil, we assessed the applicability of the HET process for remediating soil with varying moisture content, which shows that the HET process is suitable for soil

containing water up to 14% (Supplementary Note 2, Supplementary Fig. 35).”

Other minor issues:

1. The authors provides the temperature profiles of 80 V and 100 V for heavy metal removal. What is the temperature of 60 V and 120 V since you also used these voltages?

Response: We appreciate the reviewer for this suggestion. The temperature curves of 60 V and 120 V were provided (Figure R11). A higher voltage input leads to a higher temperature, but excessive voltage input (120 V) can cause inhomogeneous heating (Figure R11d). Hence, 100 V exhibits the highest heavy metals removal efficiency (Fig. 2).

Figure R11. Temperature measurement under different voltage input. Temperature profile of HET at voltages of (a) 60V, (b) 80 V, (c) 100 V, and (d) 120 V.

(This figure has been included in Supplementary Fig. 5)

2. The authors provides the temperature profile for heavy metal removal. The temperature for remediation of PAHs is also required.

Response: We appreciate the reviewer for this suggestion. The temperature profile for PAHs remediation was provided.

Figure R12. Temperature measurement for PAH remediation.

(This figure has been included in Supplementary Fig. 17)

We revised the manuscript accordingly, on P9, “**The HET conditions for PAH remediation are listed in Supplementary Table 2, with a typical maximum temperature of ~1500 °C (Supplementary Fig. 17).**”

3. The authors used a sieving process for separating carbon additives and soil, and reused the carbon additives. What's the residual carbon content in the soil after the process?

Response: We appreciate the reviewer’s comment on residual carbon. Here, we measured the soil carbon content in the raw soil as well as the treated soil following removal of the conductive carbon additives by sieving. The measurement was conducted using an elemental combustion system. It is shown that the raw soil has carbon content of ~3.7% (Figure R13). After the HET treatment and removal of carbon additive by sieving, the residual carbon content in the soil is ~3.5%. This shows that the HET process does not significantly change the soil carbon content.

Figure R13. Soil carbon content measurement. (a) Calibration curve for the soil carbon content measurement. (b) Soil carbon content in the raw soil and the treated soil after removal of the carbon additives by sieving.

(This figure has been added into SI as Supplementary Fig. 12)

We have revised the manuscript accordingly, on P8, “We measured the soil carbon content in the raw soil and the treated soil after separating carbon conductive additives (Supplementary Fig. 12). The soil carbon content in the treated soil is ~3.5%, comparable to the raw soil (~3.7%). The residual carbon additive can compensate for the organic carbon loss during the HET process, resulting in a similar total carbon content in the treated soil and raw soil.”

Method section on P24:

“Soil carbon content measurement.

The soil carbon content was measured using a ECS 4010 – CHNS-O Elemental Combustion System. Before the measurement, 1.0 g of soil sample was treated with 10 mL of 0.1 M HCl in an ultrasonic bath (Cole-Parmer Ultrasonic Cleaner) for 15 min to remove inorganic carbon (e.g., carbonate). Subsequently, the sample was dried at 105 °C to prepare for the measurement. Acetanilide was used as the standard for calibration. Both the raw soil and the treated soil, after removing carbon additives through sieving, were subjected to carbon content measurement. Analysis of samples was carried out in triplicate to afford standard deviations.”

REVIEWERS' COMMENTS

Reviewer #1 (Remarks to the Author):

Nil

Reviewer #2 (Remarks to the Author):

The authors have addressed my comments. The manuscript is in good shape to publish now.

Reviewer #1 (Remarks to the Author):

Comments:

No.

Response: We appreciate the referee for the kind review of our work.

Reviewer #2 (Remarks to the Author):

Comments:

The authors have addressed my comments. The manuscript is in good shape to publish now.

Response: We appreciate the referee for the kind review of our work.